Article 

# Entanglement in the quantum phases of an unfrustrated Rydberg atom array

**Matthew J. O'Rourke** [1] **& Garnet Kin-Lic Chan** [1] ✉

Recent experimental advances have stimulated interest in the use of large, two-dimensional arrays of Rydberg atoms as a platform for quantum information processing and to study exotic many-body quantum states. However, the native long-range interactions between the atoms complicate experimental analysis and precise theoretical understanding of these systems. Here we use new tensor network algorithms capable of including all long-range interactions to study the ground state phase diagram of Rydberg atoms in a geometrically unfrustrated square lattice array. We find a greatly altered phase diagram from earlier numerical and experimental studies, revealed by studying the phases on the bulk lattice and their analogs in experiment-sized finite arrays. We further describe a previously unknown region with a nematic phase stabilized by short-range entanglement and an order from disorder mechanism. Broadly our results yield a conceptual guide for future experiments, while our techniques provide a blueprint for converging numerical studies in other lattices.

Rydberg atom arrays consist of a set of cold neutral atoms that are trapped in an optical lattice, interacting strongly with each other via excitation into Rydberg states[1,2]. Advances in experimental control over a large number of atoms, arranged in two-dimensional arrays, have generated significant interest in using these systems for a variety of applications, including quantum information processing and stabilizing quantum states with long-range entanglement[3–17]. A recent seminal experiment[18], backed by numerical studies[19,20], has suggested a richness in the ground states of Rydberg atom arrays on a 2D square lattice. However, although the observed, non-disordered, phases are not all classical crystals, they contain little entanglement[19]. Thus it remains unclear whether such arrays realize non-trivial entangled quantum ground-states on simple lattices.

The Rydberg atom array Hamiltonian is

$$\hat{H} = \sum_{i=1}^{N} \left[ \frac{\Omega}{2} \hat{\sigma}_i^x - \delta \hat{n}_i \right] + \frac{1}{2} \sum_{i \neq j} \frac{V}{(|\mathbf{r}_i - \mathbf{r}_j|/a)^6} \hat{n}_i \hat{n}_j. \tag{1}$$

Here $\hat{\sigma}_i^x = |0_i\rangle\langle 1_i| + |1_i\rangle\langle 0_i|$ and $\hat{n}_i = |1_i\rangle\langle 1_i|$ ($\{|0_i\rangle, |1_i\rangle\}$) denote ground and Rydberg states of atom $i$). $a$ is lattice spacing, $\Omega$ labels Rabi frequency, and $\delta$ describes laser detuning. $V$ parameterizes the interaction strength between excitations. This can be re-expressed in terms of the Rydberg blockade radius $R_b$, with $V/(R_b/a)^6 \equiv \Omega$. We study the square lattice in units $a = \Omega = 1$[19], yielding two free parameters $\delta$ and $R_b$.

The ground states of this Hamiltonian are simply understood in two limits. For $\delta/\Omega \gg 1$, $R_b \neq 0$, the system is classical and one obtains classical crystals of Rydberg excitations[21–24] whose spatial density is set by the competition between $\delta$ and $R_b$. For $\delta/\Omega \ll 1$, $R_b \neq 0$, Rydberg excitations are disfavored and the solutions are dominated by Rabi oscillations, leading to a trivial disordered phase[19,25,26]. In between these limits, it is known in 1D that no other density-ordered ground states exist besides the classical-looking crystals, with a Luttinger liquid appearing on the boundary between ordered and disordered phases[26].

In 2D, however, the picture is quite different. An initial study[19] using the density matrix renormalization group (DMRG)[27–30] found additional quantum crystalline (or so-called density-ordered) phases, where the local excitation density is not close to 0 or 1. A recent experiment on a 256 programmable atom array has realized such phases[18]. However, as also discussed there, the density-ordered phases are unentangled quantum mean-field phases, and thus not very interesting. In addition, more recent numerical results[17] highlight the sensitivity of the physics to the tails of the Rydberg interaction and finite size effects. Thus, whether Rydberg atom arrays on a simple

[1]Division of Chemistry and Chemical Engineering, California Institute of Technology, Pasadena, CA 91125, USA. ✉e-mail: gkc1000@gmail.com

unfrustrated lattice—such as the square lattice—support interesting quantum ground-states, remains an open question.

Here, we resolve these questions through high-fidelity numerical simulations. To do so, we develop and employ new numerical techniques based on variational tensor network methods. Tensor networks have led to breakthroughs in the understanding of 2D quantum many-body problems[31], and our two new techniques address specific complexities of simulating interactions in Rydberg atom arrays. The first we term Γ-point DMRG, which utilizes a computational spin basis with periodic boundary conditions, and which can also be viewed as a type of DMRG that is deployed on a torus with interactions wrapping around to infinite range, while employing a traditional finite system two-dimensional DMRG methodology[27]. This removes interaction truncations and boundary effects present in earlier studies[16,17,19,20], and allows us to controllably converge the bulk phase diagram. The second is a representation of long-range interactions[32] compatible with projected entangled pair states (PEPS)[33–36]. With this, we use PEPS to find the ground states of a Hamiltonian with long-range interactions for the first time, and specifically here, model the states of finite Rydberg arrays of large widths as used in experiment. We show that, unexpectedly, the faithful inclusion of all long-range terms in our simulations yields quite different physics compared to both previous theoretical and experimental analyses. Some previously predicted ground state phases are destabilized, while other unanticipated phases emerge – including evidence of a non-trivial nematic phase stabilized by short-range entanglement in an order from disorder mechanism[37], even on the geometrically unfrustrated square lattice array. In the following, we first describe the new numerical techniques, before turning to the bulk and finite-size phase behavior of square lattice Rydberg arrays and the question of entangled quantum phases.

## Results

### Bulk simulation strategy and Γ-point DMRG

A challenge in simulating Rydberg atom arrays is the long-range tails of the interaction. Because itinerancy only arises indirectly as an effective energy scale[25], the main finite size effects arise from interactions. Many previous studies have employed a cylindrical DMRG geometry common in 2D DMRG studies[27]. However, there the interaction is truncated to the cylinder half-width, while along the open direction, edge atoms experience different interactions than in the bulk; both choices produce strong finite size effects.

To avoid these problems, we perform 2D DMRG calculations in a site Bloch basis. Given the site basis $|n_{x,y}\rangle$, $n \in \{0, 1\}$, the Bloch basis states are periodic combinations, $|\tilde{n}_{x,y}\rangle = \sum_l |n_{(x,y)+\mathbf{R}_l}\rangle$ where $\mathbf{R}_l = (n \cdot L_x, m \cdot L_y)$, $n,m \in \mathbb{Z}$, are lattice vectors, $L_x$, $L_y$ are the supercell side lengths, and $n_{x,y} = n_{(x,y)+\mathbf{R}_l}$, i.e., the occupancies at the translationally related sites are the same. The above are analogous to Bloch states at the Γ-point of the supercell Brillouin zone. The finite many-body Hilbert space under the Γ-point restriction is $\prod_{x,y}|\tilde{n}_{x,y}\rangle$; this Hilbert space should be interpreted as a model of the Hilbert space of the infinite system, rather than a true subspace of it. The corresponding matrix product state (MPS) is expressed as $|\Psi\rangle = \sum_{\{e\}} \prod_{x,y} A^{\tilde{n}_{x,y}}_{\{e_{x,y}\}}|\tilde{n}_{x,y}\rangle$ where $\mathbf{A}^{\tilde{n}_{x,y}}$ is the MPS tensor associated with Bloch function $\tilde{n}_{x,y}$, $e_{x,y}$ denote its bonds, and a 2D snake ordering has been chosen through the lattice. In the above picture, Γ-point 2D DMRG formally models an infinite lattice (Fig. 1a) with a wavefunction constrained by the supercell shape. This differs from using a periodic MPS as periodicity is enforced by the Bloch basis rather than the MPS. The Γ-point DMRG calculation can also be viewed as working on a finite toroidal geometry (i.e., the supercell) with the typical pure site basis $|n_{x,y}\rangle$, but where the interactions are allowed to wrap infinitely around the torus, rather than being cut off. In either interpretation, the Hamiltonian per supercell contains interactions expressed as an infinite lattice sum,

$$\hat{H} = \sum_i \left[\frac{1}{2}\hat{\sigma}_i^x - \delta\hat{n}_i\right] + \frac{1}{2}\sum_{i \neq j + \mathbf{R}_l, \mathbf{R}_l} \frac{R_b^6}{|\mathbf{r}_i - \mathbf{r}_{j+\mathbf{R}_l}|^6}\hat{n}_i\hat{n}_j, \quad (2)$$

Further details of this approach and its two interpretations are discussed in the Methods section.

The only finite size parameter is the supercell size $L_x \times L_y$. We thus perform exhaustive scans over $L_x, L_y$ to identify competing ground state orders. We systematically converge the energy per site of low-

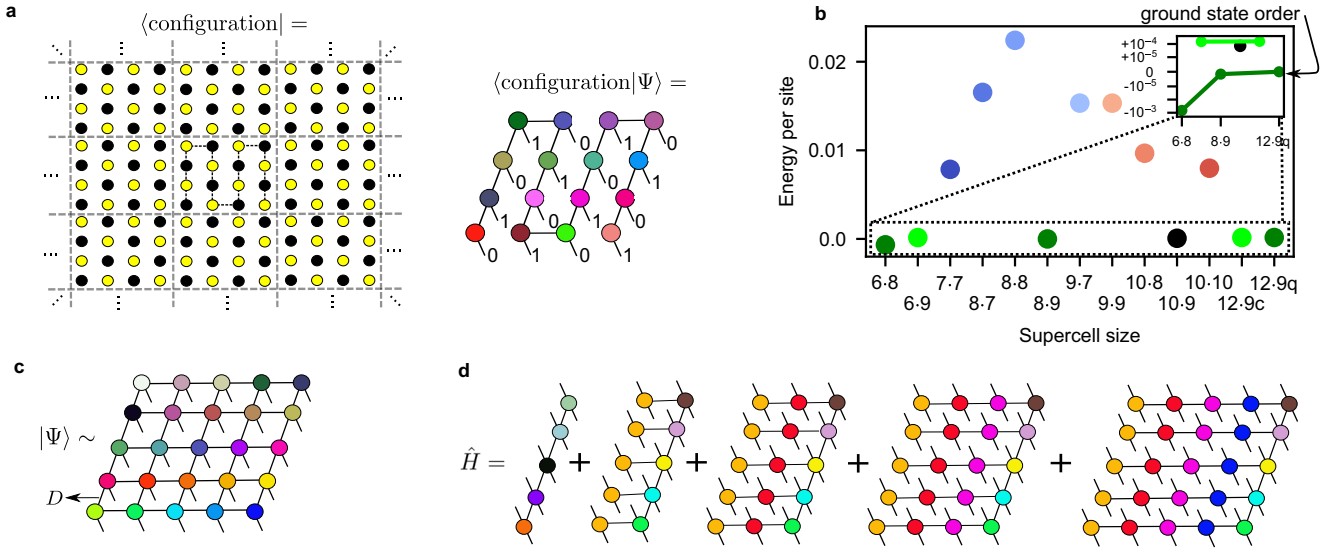

Fig. 1 | **Numerical methods and strategy. a** A schematic representation of Γ-point DMRG. A single infinite bulk configuration is given by periodic images of the central supercell configuration. The wavefunction coefficient for this infinite configuration is given by the contraction of a snake MPS, which is defined only within a single supercell. **b** By widely varying the size of the supercell, Γ-point DMRG obtains many different ground states. Identifying all accessible supercells which give the same ground state order (shown with identically colored points), we can ensure that all competing low-energy states are well converged w.r.t. finite size effects, and thus

properly identify the true ground state (inset shows ground-state order (dark green) converged w.r.t. supercell size, separated from other low-energy orders by $10^{-4}$ energy units). **c** A PEPS wavefunction ansatz with bond dimension $D$ for a finite system. Each tensor is a different color because they can all be unique. **d** A simplified diagrammatic representation of the long-range Hamiltonian construction for PEPS in ref. 32. All terms in the Hamiltonian are accounted for by a sum of $L_x$ comb tensor network operators. Tensors of the same color are identical.

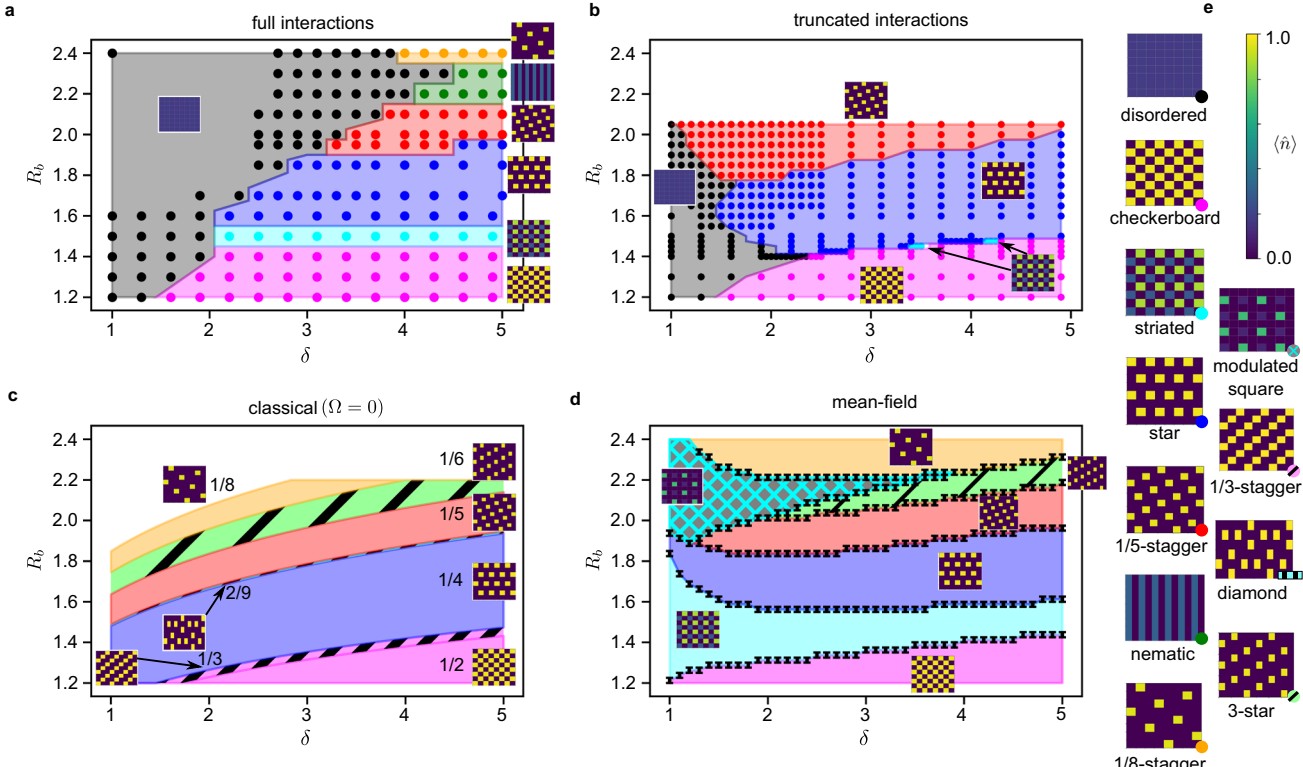

**Fig. 2 | Phase diagrams of the bulk system under various assumptions.** The color of a dot/region identifies the ground state order. The density profiles for each color are given in (**e**) and shown near each phase domain. **a** The phase diagram given by Γ-point DMRG including all long-range interactions. **b** The phase diagram from Γ-point DMRG when interactions are truncated to 0 beyond a distance of $|\mathbf{r}_i - \mathbf{r}_j| = 2$. **c** The classical phase diagram (when all sites are either fully occupied or empty) including all long-range interactions. **d** The mean-field phase diagram, including all long-range interactions. Error bars display the uncertainty of the computed phase boundaries. **e** Representative density profiles for all phases in (**a–d**), identified by the colored dot in each lower right corner. All profiles have Γ-point boundary conditions on all edges. In (**a**, **b**) dots denote computed data, while shading is a guide for the eye. (**c**, **d**) are computed with very fine resolution/analytically, thus no dots are shown.

energy orders by increasing the commensurate supercell sizes to contain many copies of the order (up to 108 sites). The finite size effects converge rapidly because no interactions are truncated and there are no edge effects even in the smallest cells, allowing us to converge the energy per site to better than $10^{-5}$, compared to the smallest energy density difference we observe between competing phases of ~$10^{-4}$ (see Fig. 1b and Supplementary Methods).

**Finite simulations and PEPS with long-range interactions**

To simulate ground-states of finite arrays, we consider finite systems (with open boundaries) of sizes $9 \times 9$ up to $16 \times 16$ atoms. This resembles capabilities of near-term experiments[10,18]. The width of the largest arrays challenges what can be confidently described with MPS and DMRG for more entangled states. Consequently, we employ PEPS wavefunctions which capture area law entanglement in 2D, and can thus be scaled to very wide arrays (Fig. 1c). Together with DMRG calculations on moderate width finite lattices, the two methods provide complementary approaches to competing phases and consistency between the two provides strong confirmation. However, PEPS are usually combined with short-range Hamiltonians. We now discuss a way to combine long-range Hamiltonians efficiently with PEPS without truncations.

For this, we rely on the representation we introduced in ref. 32. This encodes the long-range Hamiltonian as a sum of comb tensor network operators (Fig. 1d). As discussed in ref. 32, arbitrary isotropic interactions can be efficiently represented in this form, which mimics the desired potential via a sum of Gaussians, i.e. $\frac{1}{r^6} = \sum_{k=1}^{k_{max}} c_k e^{-b_k r^2}$ (where $k_{max} \sim 7$ for the desired accuracy in this work). The combs can

be efficiently contracted much more cheaply than using a general tensor network operator.

While ref. 32 described the Hamiltonian encoding, here we must also find the ground-state. We variationally minimize $\langle \Psi | \hat{H} | \Psi \rangle$ using automatic differentiation[38]. Combined with the comb-based energy evaluation, this allows for both the PEPS energy and gradient to be evaluated with a cost linear in lattice size. Further details are discussed in the Methods section, including some challenges in stably converging the PEPS optimization.

**Summary of the bulk phase diagram**

Figure 2a shows the bulk phase diagram from Γ-point DMRG with infinite-range interactions. We first discuss the orders identified by their density profiles (orders of some phase transitions are briefly discussed in Supplementary Note 3). Where we observe the same phases as in earlier work[19], we use the same names, although there are very substantial differences with earlier phase diagrams.

With weaker interactions ($R_b < 1.8$), the ground states progress through densely-packed, density-ordered phases starting from checkerboard (pink, $R_b \sim 1.2$), to striated (cyan, $R_b \sim 1.5$), to star (blue, $R_b \sim 1.6$). While the checkerboard and star phases are classical-like crystals, the striated state is a density-ordered quantum phase, seen previously[19].

With stronger interactions ($R_b > 1.8$), the phases look very different from earlier work, which truncated the interactions[19]. Ordered ground states start with the $\frac{1}{5}$-staggered phase (red, $R_b \sim 1.95$), then progress to a nematic phase (dark green, $R_b \sim 2.2$) and the $\frac{1}{8}$-staggered phase (gold, $R_b \sim 2.4$). There is also a small region at larger $\delta$ (not

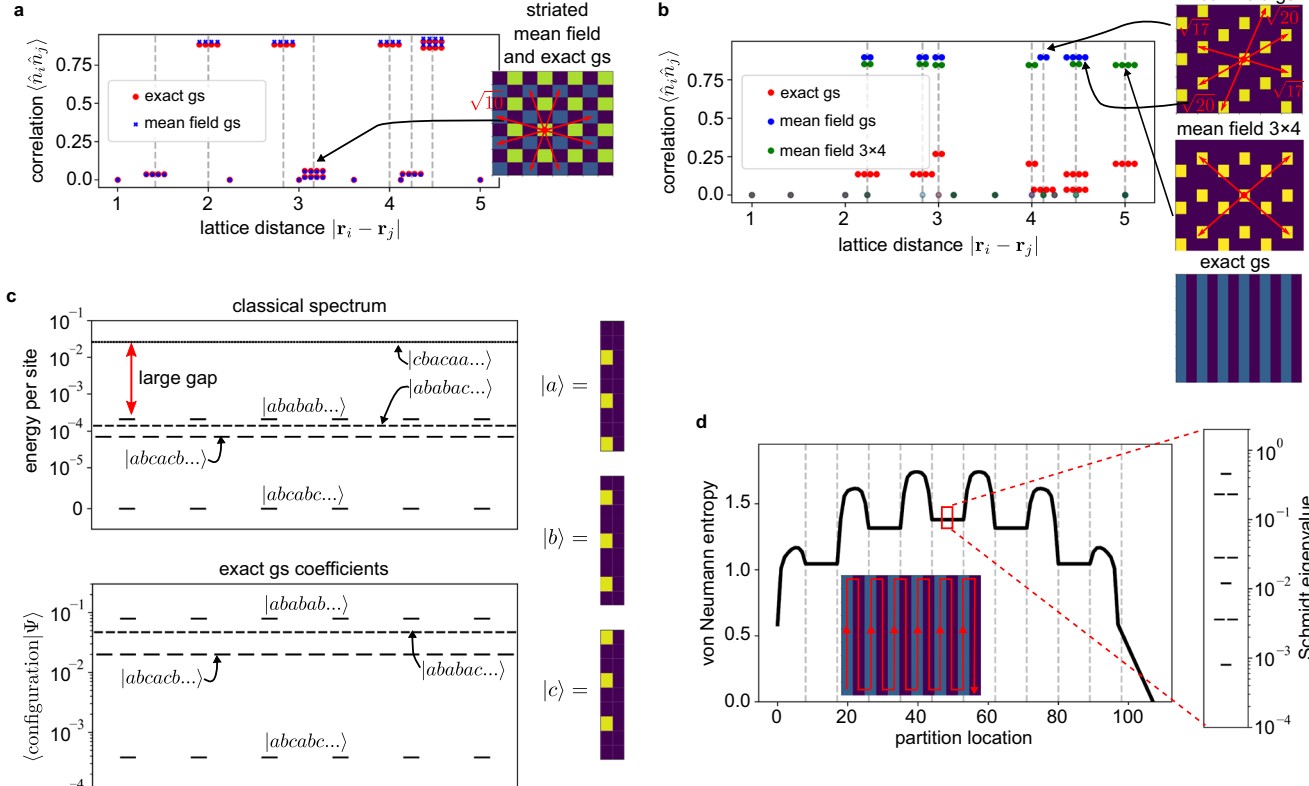

**Fig. 3 | Mean-field striated versus entangled nematic phase. a** Density-density correlation functions of the mean-field and exact striated ground state, both at $(\delta, R_b) = (3.1, 1.5)$; these agree, confirming the mean-field nature of the striated phase. (b) Density-density correlation functions for the entangled nematic phase ground state and two different mean-field ground states (from a 6 × 3 unit cell and a 3 × 4 unit cell) at $(\delta, R_b) = (5.0, 2.3)$. In (**a, b**), 2-fold/4-fold degeneracy of a peak is indicated by 2/4 horizontal dots distributed around the proper distance coordinate. 8-fold degeneracy in (**a**) is shown as two rows of 4 dots. The non-mean-field (entangled) character of the nematic phase is evident. **c** Structure of the nematic state in terms of classical configurations constructed via compositions of 3 individual column states $|a\rangle, |b\rangle, |c\rangle$. In the classical limit, there are 4 distinct sets of low-energy configurations, all characterized by the absence of adjacent columns in the

same state (e.g., $|aa...\rangle$) and large degeneracies due to permutational symmetry between $|a\rangle$, $|b\rangle$, and $|c\rangle$. The lowest in energy is 6-fold degenerate, corresponding to the 3-star state. However, in the quantum nematic state the configurations that are slightly higher in energy have much larger wavefunction coefficients. The most relevant classical states in the wavefunction are those with the greatest number of possible single full column hops (e.g., $a \to b$) without introducing unfavorable states like $|aa...\rangle$, revealing the role of itinerancy in the nematic phase. **d** Bipartite entanglement entropy for each possible bipartition of the 12 × 9 supercell nematic ground state. One inset shows the path that the partition location axis follows through the supercell MPS, while the other shows the entanglement spectrum at a central cut.

shown) where the nematic phase and a classical-like crystal (which we call 3-star) appear to be essentially degenerate, with an energy difference per site of $\Delta e < 3 \cdot 10^{-5}$ (see Supplementary Note 2).

### Effects of interactions on the bulk phases

In Figure 2b, we show the phase diagram computed using Γ-point DMRG with interactions truncated to distance 2. This approximation resembles earlier numerical studies[19], but here bulk boundary conditions are enforced by the Bloch basis, rather than cylindrical DMRG. Comparing Fig. 2a, b, we see the disordered and striated phases are greatly stabilized using the full interaction, and new longer-range orders are stabilized at larger $R_b$. Comparing Fig. 2b and ref. 19, we see that having all atoms interact on an equal footing (via the Bloch basis) destroys some quantum ordered phases seen in ref. 19 at larger $R_b$.

### Classical, mean-field, and entangled bulk phases

Without the Rabi term $\Omega$, one would obtain classical Rydberg crystals without a disordered phase. Figure 2c shows the classical phase diagram. For the $\delta$ values here, the 1D classical phase diagram has sizable regions of stability for all accessible unit fraction densities[22,26]. However, the connectivity of the square lattice in 2D

changes this. For example, only a tiny part of the phase diagram supports a $\frac{1}{3}$-density crystal, and we do not find a stable $\frac{1}{7}$-density crystal within unit cell sizes of up to 10 × 10. All ordered quantum phases in Fig. 2a appear as classical phases except for the striated and nematic phases, while there are small regions of classical phases at densities $\frac{1}{3}$ and $\frac{2}{9}$ with no quantum counterpart. The striated and nematic phases emerge near the $\frac{1}{3}$ and $\frac{1}{7}$ density gaps respectively, however the nematic phase also supersedes the large region of the $\frac{1}{6}$ density 3-star crystal.

Ref. 18 suggested that quantum density-ordered phases are qualitatively mean-field states of the form $\prod_i \alpha_i |0_i\rangle + \sqrt{1 - |\alpha_i|^2}|1_i\rangle$. Figure 2d shows the mean-field phase diagram. The disordered phase does not appear, as it emerges from defect hopping and cannot be described without some entanglement[25]. The mean-field phase diagram contains features of both the classical and quantum phase diagrams. The striated quantum phase indeed appears as a mean-field state, confirmed by the match between the mean-field and exact correlation functions (Fig. 3a). However, the nematic phase does not appear, and in its place is the same $\frac{1}{6}$-density crystal stabilized in the classical phase diagram. This shows that a treatment of entanglement is required to describe the nematic phase.

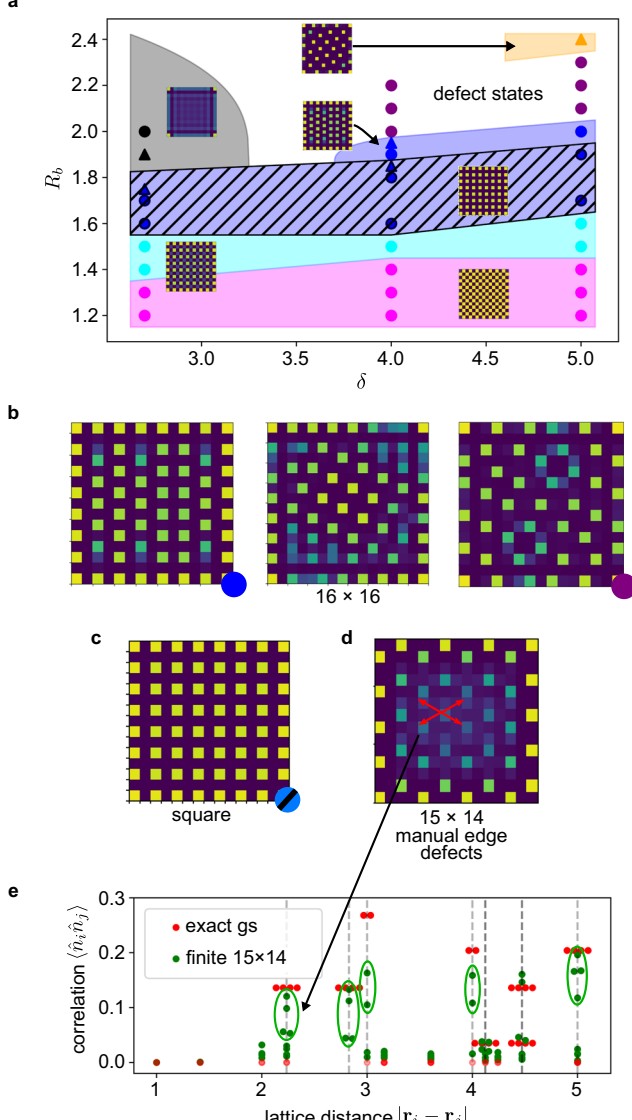

**Fig. 4 | Phase diagram of the 15 × 15 finite system and finite lattice orders. a** The phase diagram, where colors correspond to the same phase classifications as Fig. 2. Triangles represent tentative classification of points showing inconsistent PEPS convergence, see Supplementary Methods. A new order, which we call square, is specified in (**c**) and various examples of boundary-bulk frustrated ground states in (**b**). **d** The density profile for a nematic-like ground state that can be stabilized on a 15 × 14 lattice at $(\delta, R_b) = (3.4, 2.1)$ with manually tailored edge excitations (see text). **e** Comparing the correlations of the finite nematic phase to the converged bulk phase. The degeneracy of the peaks is split by the boundary excitations, but the number of peaks is generally conserved between the two (green ovals), which provides a clear distinction from mean-field states (see Fig. 3b).

## Nature of the bulk nematic phase

Figure 3b shows the density correlation function of the nematic phase, which does not display mean-field character. To reveal the phase structure, Fig. 3c shows the lowest energy classical states in the same region of the phase diagram. Due to the Rydberg blockade radius ($R_b = 2.3$), excitations are spaced by 3 units within a column, giving 3 column configurations $|a\rangle$, $|b\rangle$, $|c\rangle$. Column-column interactions, however, prevent adjacent columns from being in the same configuration (with excitations separated by 2 units); thus, states such as $|abcb\ldots\rangle$ are allowed, but $|accb\ldots\rangle$ are not. Without long-range interactions, these column constraints give rise to an exponential classical degeneracy. Long-range interactions partially lift the classical

degeneracy, yielding the $|abc\ldots\rangle$ crystal (3-star phase) and its 6-fold degenerate permutations. However, after including quantum fluctuations and entanglement through a 4th order perturbative treatment of $\sigma_x$ (giving rise to defect itinerancy), $|abab\ldots\rangle$ and related configuration energies are lowered below those of the $|abc\ldots\rangle$ configurations; the fluctuations stabilize non-classical crystal configurations (see Supplementary Note 1). Figure 3c gives the weights of the configurations in the computed quantum ground-state, which are distributed across the exponentially numerous non-classical $|abab\ldots\rangle$, $|abcbab\rangle$ etc., configurations, with the classical crystal $|abc\ldots\rangle$ configurations strongly disfavored. The bi-partite entanglement entropy and entanglement spectrum are further shown in Fig. 3d. Although the fluctuations are presumably of finite range, the entanglement spectrum carries 3 large Schmidt values across every cut along the DMRG snake MPS, showing the state is entangled across the entire supercell, and well approximated by an MPS of bond dimension 3. The entanglement structure emerges from the combination of defect itinerancy and the constraints on adjacent columns. Thus, it is clear that quantum fluctuations are much stronger in this phase than in any of the surrounding ordered phases. Assuming the entanglement is ultimately short-ranged (i.e., on scales beyond the supercell sizes we can treat here), this phase can be identified as containing strong fluctuations around a non-classical crystal, stabilized by an order from disorder mechanism[37] (further discussion in Supplementary Note 1).

## Finite phase diagram

Current experiments are limited to lattices with open boundary conditions consisting of a few hundred atoms[10,18]. To investigate how this modifies the bulk behavior, we computed the phase diagram of selected finite lattices from size 9 × 9 to 16 × 16, using DMRG for the smaller sizes and our PEPS methodology for the larger ones.

We first focus (in Fig. 4a) on understanding the fate of the ordered phases on the 15 × 15 lattice along three slices: $\delta = 2.7$, 4.0, and 5.0 (16 × 16 lattice phases, as well as other lattice sizes, are discussed in Supplementary Notes 4–5). Here, many finite lattice ground state orders resemble those in the bulk. However, their regions of stability are substantially reduced and their patterns are broken by frustration. Out of the density-ordered quantum phases, the striated mean-field phase remains due to its commensurate boundary-bulk configurations, while in the region of strongest interactions, the nematic phase is destabilized. A new region of classical order, called here the square phase (Fig. 4c), emerges across much of the $R_b = 1.5$–1.8 region where the star phase was stable in the bulk[20]. We distinguish the square order from the striated order in the sense that the former has negligible quantum fluctuations on the (1, 1)-sublattice, although it is unclear if the square and striated orders constitute truly distinct phases (in the bulk phases the square order is not stable, only the striated order appears).

In Fig. 5, we directly compare the experimental results on the 13 × 13 lattice to our calculations on the same lattice. The analysis of the experiments in ref. 18 was based on simulations on the 9 × 9 lattice using truncated interactions. This assigned only part of the experimental non-zero order parameter space to a square/striated phase (see Fig. 5a, note, the order parameter does not distinguish between square/striated orders). However, our simulations (Fig. 5c) in fact reproduce the full region of the non-zero order parameter, and thus, the whole region seen experimentally should be assigned to a square/striated phase, with the square order appearing in the upper part of the region. Similarly, the experimental analysis identified a large region of star order (Fig. 5b). This assignment is complicated by edge effects, which mean that the order parameter used does not cleanly distinguish the star phase from other phases. However, our simulations suggest that the region of the star phase should be considered to be much smaller, located at the very top of the non-zero order region, and this is confirmed using a different, more sensitive order parameter

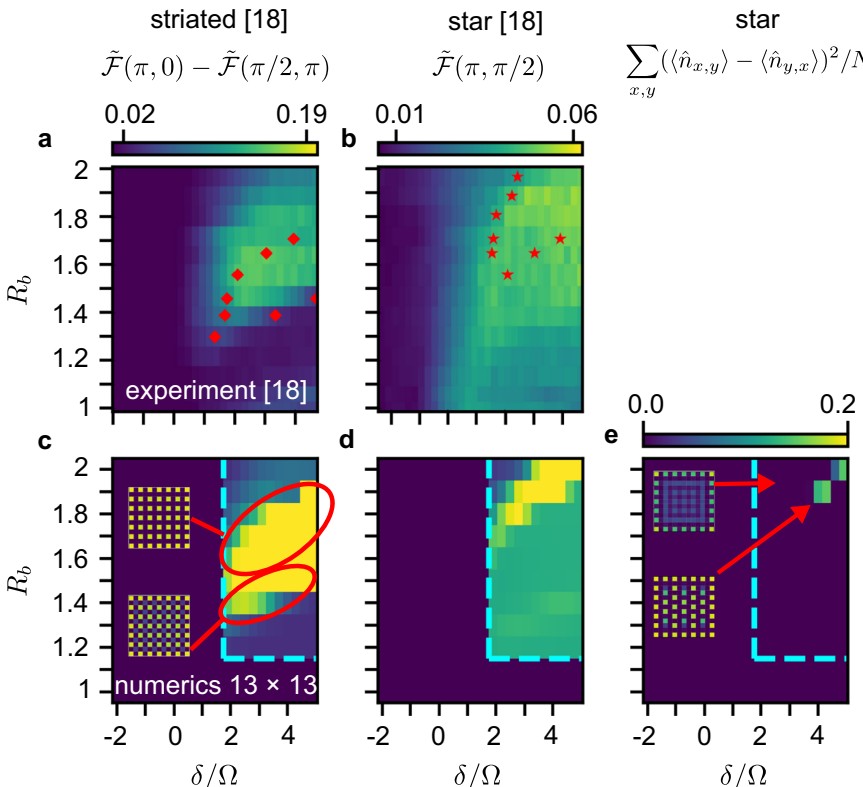

**Fig. 5 | Comparison to experiment.** The (**a**, **b**) row directly reproduces experimental phase diagram data on the $13 \times 13$ lattice (data extracted from ref. 18 Fig. 4), while the (**c**–**e**) row is $13 \times 13$ numerical data computed in this work. The first two columns show the order parameters used in ref. 18 to identify the square/striated and star phases, while the third column shows a new, more sensitive order parameter for the star phase. Red dots in (a)-(b) denote the phase boundaries assigned in ref. 18, while the cyan dotted lines in (c–e) indicate the subset of parameter space that was computed. Our calculations support a re-interpretation of the experimental data with a significantly larger square/striated region and much smaller star phase.

(Fig. 5e). Overall, the measured data corresponds more closely to our numerics than earlier simulations, giving confidence in our more precise interpretation (more discussion in Supplementary Note 5).

### Stabilizing the finite analog of the nematic order
Generally, the impact of boundary physics can be understood in terms of frustration of the bulk order by the boundary order, where excitations concentrate more densely due to the lower energetic penalty from fewer long-range interactions on the edge. Examples of the effects of this frustration, ranging from modified bulk orders, to defect dominated states, to boundary-only orders are shown in Figs. 4b, c and 5e (see also Supplementary Note 4).

We searched for conditions to stabilize the nematic ground-state on a finite lattice by manipulating boundary effects. We scanned various rectangular sizes and explicitly removed patterns of atoms from the edges to induce different bulk orders. We found the best conditions to stabilize a finite-size analogy of the nematic phase occur near $(\delta, R_b) = (3.4, 2.1)$, on a $15 \times 14$ lattice, while removing edge atoms to create a spacing of 4 on two edges and 3 on the other two edges (Fig. 4d). Note that the location of this state in phase space cannot be directly compared to the locations of states in Fig. 4a due to the significant difference in the treatment of the boundary. Although there are strong finite size effects, the density profile and correlation functions (Fig. 4d, e) reveal qualitative similarities to the bulk nematic phase, in particular, the presence of 4-fold correlation peaks at distance $\sqrt{5}$ and $\sqrt{8}$, which are also a feature of the bulk entangled phase (Fig. 3b). Importantly, the multiplicity of these peaks would be different in the classical or mean-field ground-states at this density.

### Discussion
Using new tensor network simulation methods, we have obtained a converged understanding of the phase diagram of Rydberg atom arrays in both bulk and finite simple square lattices. Surprisingly, our bulk phase diagram is quite different from that predicted in earlier numerical studies, while on finite lattices, our results support a reinterpretation of previous experimental analysis. Theoretically, this is due to the subtle effects of the long-range interactions that are addressed by our techniques, while experimentally, it brings into focus the challenge of more accurate theoretical models to interpret increasing experimental capabilities in quantum many-body physics. Perhaps most intriguingly, we find strong evidence that the geometrically unfrustrated square lattice supports a nematic phase with strong fluctuations, stabilized by an order from disorder mechanism involving the competition between emergent itinerancy and the constraints of the Rydberg interaction.

A primary focus of Rydberg atom array experiments has been to realize well-studied short-range Hamiltonians, for example, on frustrated lattices. However, we find that lattice frustration is not necessary to produce interesting entanglement effects in Rydberg systems. In fact, our work highlights the richness and complexity intrinsic to Rydberg atom arrays, due to the non-trivial effects of their native interactions.

### Methods
A brief conceptual discussion of our new numerical techniques was already presented in the Results section. Here, we will focus on more algorithmic details and subtleties.

## Γ-point DMRG: theory and relation to other methods

In this work we chose to perform 2D DMRG in a site Bloch basis at the Γ-point in the Brillouin zone. Let us define the computational supercell of the DMRG calculation to be of dimension $L_x \times L_y$ sites. Then, such a Γ-point site Bloch basis state $|\tilde{n}_{x,y}\rangle$ is related to the normal site basis state $|n_{x,y}\rangle$ at site $r_{x,y}$ by

$$|\tilde{n}_{x,y}\rangle = \sum_l |n_{(x,y)+\mathbf{R}_l}\rangle \tag{3}$$

where $\mathbf{R}_l = (n \cdot L_x, m \cdot L_y)$; $n, m \in \mathbb{Z}$. In other words, each single particle basis state is a superposition of the original site basis states separated by lattice vectors of the supercell. The occupancies of sites related by the supercell lattice vectors, i.e., $n_{x,y}$ and $n_{(x,y)+\mathbf{R}_l}$ are constrained to be the same. The Bloch function has unit norm per supercell.

The many-particle 2D DMRG wavefunction is then

$$|\Psi\rangle = \sum_{\{e\}} \prod_{x,y} A^{\tilde{n}_{x,y}}_{\{e_{x,y}\}} |\tilde{n}_{x,y}\rangle \tag{4}$$

where $\mathbf{A}^{\tilde{n}_{x,y}}$ is the MPS tensor associated with Bloch function $\tilde{n}_{x,y}$, $e_{x,y}$ denote its bonds, and a standard snake ordering has been chosen through the lattice[27]. The Hilbert space is $\prod_{x,y}|\tilde{n}_{x,y}\rangle$, i.e., it is of dimension $2^{L_x L_y}$. Note that, for supercells larger than a single site, the Hilbert space is a product of Bloch functions, but no double occupancy occurs because different Bloch functions $\tilde{n}_{x,y}$ occupy non-overlapping sites on the infinite lattice. The final Hilbert space is best viewed as a model of the Hilbert space of the infinite system, rather than a subspace in the Hilbert space of the infinite system. We have implemented this strategy with the ITensor software library[39].

As mentioned earlier, this representation is different from the cylindrical boundary condition MPS employed in previous studies[16,17,19]. The primary advantage of the current approach is that regardless of supercell size, the 2D DMRG state models an infinite system in 2D (rather than a finite system in at least one direction in prior cylindrical studies) simply because the underlying single-particle basis is a discrete periodic function on the infinite 2D square lattice. Thus there is no need to truncate the Rydberg interactions unlike in cylinder studies. We note that this type of Bloch basis, i.e., linear combinations of local states separated by (supercell) lattice vectors, is widely used in electronic structure theory partly for similar reasons, namely, it allows one to treat the infinite range Coulomb interaction. For an example of a DMRG calculation of an infinite system using such Bloch bases (known as crystalline atomic orbitals) in electronic structure, see e.g., ref. 40.

Systematic convergence to the correct bulk behavior in the Bloch representation is controlled by two parameters: the DMRG bond dimension and the size of the supercell. The Γ-point basis functions for larger supercells span larger and larger models of the Hilbert space of the infinite system. Examining convergence with bond dimension and supercell size is fully sufficient to establish convergence to the thermodynamic limit. Because of the hardcore constraints of the bosons, it is not convenient to consider the product space of Bloch states at different points in the supercell Brillouin zone. However, we could in principle choose to define *all* Bloch states in Eq. (3) to be away from the Γ point in the supercell Brillouin zone, equivalent to adding phase factors in Eq. (3). This would correspond to a twisted boundary condition, and averaging over such boundary conditions might be expected to further reduce finite size effects.

One way to understand the 2D DMRG calculation in the Bloch basis is to examine the form of the correlation functions it predicts for an infinite system. Because the Bloch states at the Γ-point are periodic, all correlation functions are implicitly periodic across supercells. For example, transformed to the site basis, the density-density correlation

function satisfies

$$\left\langle n_{x_1,y_1} n_{x_2,y_2} \right\rangle = \left\langle n_{x_1,y_1} n_{(x_2,y_2)+\mathbf{R}_l} \right\rangle. \tag{5}$$

Particles in adjacent supercells are thus entangled and correlated with each other, but in a highly constrained fashion. (This can be seen from the entanglement of a single particle state in the Bloch basis, which has the maximum entanglement entropy of log 2 for a cut in the site basis). Note that a 2D infinite tensor network, such as an iPEPS, also introduces a constrained form of correlations between particles; but the constraint there is different and controlled solely by the bond dimension. In the 2D DMRG calculations in the Bloch basis, the full flexibility of long-range correlations is restored by increasing the supercell size.

An alternative, and completely equivalent, way to describe the 2D DMRG calculation in the Bloch representation at the Γ point is to map it to a calculation on a finite system. This finite system is a torus of dimension $L_x \times L_y$; we see that it has a Hilbert space of the same dimension, labeled by the same occupancies $|\tilde{n}_{x,y}\rangle$; thus the model Hilbert space of the infinite system at the Γ point can be identified with the toroidal Hilbert space.

In the Γ-point picture, the transformation from the basis $|n_{x,y}\rangle$ to the Bloch basis $|\tilde{n}_{x,y}\rangle$ modifies the interaction from the original Rydberg form to an infinite lattice sum over the real space lattice (Eq. (2)). This Hamiltonian (Eq. (2)) then encodes the per supercell energy of the infinite bulk system. In the toroidal picture, this lattice sum can be viewed as arising from taking interactions that loop around a torus infinitely many times, with the proper decaying form. There is a 1-1 mapping between the toroidal representation with infinite wrap-around, and the Γ-point supercell formulation discussed above. For example, the torus representation can also be generalized to the twisted Bloch basis discussed above: this corresponds to inserting hoppings across the torus with a phase factor. Which language is used is thus primarily a matter of preference.

Further details regarding analysis of finite size errors and convergence strategy can be found in Supplementary Methods.

## PEPS: overview

The PEPS simulations in this work combine recent advances in optimizing PEPS wavefunctions using automatic differentiation[38] and 2D operator representations of long-range interactions[32]. This combination illuminated many new challenges for PEPS optimization with respect to complicated Hamiltonians. The following sections will detail the various challenges and the technical solutions used in this work. The instability of PEPS optimization remains an open problem and it is an area of future research to determine a PEPS optimization pipeline (using automatic differentiation) that is fully robust to problem instance. In the following sections, $D$ will refer to the PEPS bond dimension and $\chi$ will refer to the maximum bond dimension allowed during contraction before approximations (via singular value decomposition, SVD) are performed. The algorithms were implemented with the quimb software package[41], using PyTorch as the backend library for automatic differentiaion[42].

## PEPS: operator representation

The method proposed in ref. 32 to represent Hamiltonians with long-range interactions writes the interaction potential as a sum of Gaussians,

$$\frac{1}{\left(\sqrt{x^2+y^2}\right)^6} \approx \sum_{k=1}^{K} c_k e^{-\lambda_k(x^2+y^2)} \equiv V_{\text{fit}}(\mathbf{r}). \tag{6}$$

Using standard methods for fitting functions by exponential sums[43,44], we can obtain a $K = 7$ fit with error $\epsilon = \max_i |1/\mathbf{r}_i^6 - V_{\text{fit}}(\mathbf{r}_i)| = 10^{-5}$ on the domain $\mathbf{r} \in [1, 16\sqrt{2}]$, which is used throughout the work.

## PEPS: essential computational techniques

As originally discussed in ref. 38, when trying to use automatic differentiation to optimize a PEPS there are a few essential techniques that must be employed, which are not typically default in standard automatic differentiation libraries. They are essential; without them the computation of the energy expectation value and its derivative will typically not run to completion due to out-of-memory errors or numerical infinities.

The first techniques is numerical stabilization of the gradient of SVD, by adding Lorentzian broadening to the inverse singular values. Consider a standard SVD of a rectangular matrix $A = USV^T$. In reverse-mode automatic differentiation, the derivative of this operation is given by,

$$\overline{A} = \frac{1}{2} U \left[ F_+ \odot \left( U^T \overline{U} - \overline{U}^T U \right) + F_- \odot \left( V^T \overline{V} - \overline{V}^T V \right) \right] V^T + U \overline{S} V^T + (I - UU^T) \overline{U} S^{-1} V^T + U S^{-1} \overline{V}^T (I - VV^T). \tag{7}$$

Here, $\overline{U}$, $\overline{S}$, and $\overline{V}$ are the derivatives (or, adjoints) of $U$, $S$, and $V$ with respect to the preceding operations in the reverse-computational graph. $[F_\pm]_{ij} = \frac{1}{s_j - s_i} \pm \frac{1}{s_j + s_i}$ for $i \neq j$, otherwise $F = 0$, where the $s_i$ are individual singular values. In the case of (quasi-) degeneracy of singular values, or if their magnitudes becomes vanishing small, $\overline{A}$ is not well-defined. At the cost of introducing a small error into the gradients, this issue can be practically resolved by applying a Lorentzian broadening to the various inverses, e.g., $\frac{1}{s_j - s_i} \approx \frac{s_j - s_i}{(s_j - s_i)^2 + \epsilon}$. In this work we use $\epsilon = 10^{-11}$.

The second essential technique is the broad usage of intermediate checkpointing when evaluating the energy to reduce the memory load of computing gradients. This is a well-known technique in reverse-mode automatic differentiation that trades additional compute time for a lower peak memory usage. Consider the forward-pass computational graph to evaluate the energy. After every $n$ steps in the graph, one can save an intermediate of the computation and discard all the other intermediates within the $n$-step interval that automatic differentiation libraries would typically need to store. Then, to propagate through the reverse-pass computation graph (to compute the gradients), a single $n$-step chunk is run in forward-pass to populate all the necessary intermediates in that segment of the graph. The reverse-mode computation can then progress through that segment, and the process is repeated for the subsequent $n$-step segments until the entire reverse-graph has been computed. The key for application with PEPS is to choose the proper intermediates to store, which do not require too much memory (i.e., store intermediates after compressing their bond dimensions).

## PEPS: stabilizing the optimization

A straightforward implementation of the energy expectation value as described in ref. 32, with optimization via automatic differentiation including the above techniques, typically fails to find the ground state PEPS for the Rydberg Hamiltonian (see Supplementary Fig. 3). This failure can be generally attributed to the fact that in the quantity under optimization $E = \frac{\langle \psi | H | \psi \rangle}{\langle \psi | \psi \rangle}$, both the numerator and denominator are evaluated approximately and thus the computation is not strictly bound by the variational principle. Consequently, the optimization can find pathological regions of the PEPS parameter values which make the PEPS contractions inaccurate for the chosen $\chi$, even when starting from an accurately contractible PEPS. Unfortunately, in this problem, we find that simply raising the value of $\chi$ does not prevent this behavior until $\chi$ is impractically large.

In order to mitigate this problem, we use the following four techniques in tandem:

- We employ line search methods that minimize the gradient norm as well as the energy. In this work, we use the BFGS algorithm[45] in conjunction with such a line search, as suggested in ref. 38.

- We use the cost function $E_1/2 + E_2 + \lambda |E_2 - E_1|$ where $E_1$ and $E_2$ are the energies of PEPS on lattices rotated by 180 degrees and $\lambda$ is a penalty factor. This strongly penalizes the optimization from entering parameter space with large contraction error (where $E_1$ and $E_2$ would be very different).

- During the first iterations of the gradient optimization we only update small patches of tensors at a time, which are chosen to break spatial symmetries that may be contained in the initial guess. After this has pushed the optimization towards the symmetries of the true ground state order, then all tensors can be updated at each optimization step.

- We evaluate the numerator and denominator of $E$ in a consistent way by using a technique we call local normalization. During the computation of $\langle \psi | H | \psi \rangle$, writing $H$ as a comb tensor sum $H = \sum_{i=1}^{L_x} h_i$, then for each comb tensor numerator $\langle \psi | h_i | \psi \rangle$, the associated denominator uses the identical contraction, but with $h_i$ replaced by the identity (the environments are not recomputed).

Combining all four of these techniques removes the most egregious instabilities in the optimization trajectory (see Supplementary Fig. 3), at the cost of a slightly larger computational burden. However, as in more standard DMRG calculations with small bond dimension, convergence to the correct ground-state (rather than a local minimum) still requires a reasonable initial guess.

## PEPS: initial guess

Obtaining an accurate ground state PEPS typically relies on starting with an accurate initial guess. The predominant algorithms to generate such a guess for problems with a local Hamiltonian are simple update[46–48] or imaginary time projection of a converged small $D$ solution to a larger $D$ guess. However, in the presence of long-range interactions it becomes challenging to generalize either of these methods in an efficient and/or accurate way. We, therefore, used the following simple scheme to generate initial guesses in this work.

- Sum $n$ manually constructed $D = 1$ PEPS to obtain an initial PEPS of bond dimension $D = n$. The configurations of these $D = 1$ PEPS were set to reproduce specific low energy Rydberg crystals and defects within them.

- For small $R_b$: truncate the long-range interactions in $H$ to next-nearest, or next-next-nearest, neighbor interactions (distance of $\sqrt{2}$ or 2), and run conventional simple update starting from the above manually summed PEPS. This fails once the ground state excitations are spaced by more than 2.

- For large $R_b$: add positive random noise to the manually summed PEPS, and then run a highly approximate, first-order gradient optimization for ~25 iterations using a large step size when updating the parameters.

Further details regarding the convergence can be found in Supplementary Methods.

## Finite 2D DMRG

Standard 2D DMRG calculations with open boundaries were used to study the $9 \times 9$ system, a low-entanglement region of the $13 \times 13$ system, and to supplement convergence of PEPS on the larger $15 \times 14$, $15 \times 15$, and $16 \times 16$ lattices. Like the PEPS calculations, these too included all long-range interactions (according to Eq. (1)). The maximal bond dimension used for the $9 \times 9$ and $13 \times 13$ simulations was $D_{\max} = 1200$, which we found was more than enough to accurately study the regions of interest in Fig. 5 for these lattices (see Supplementary Fig. 6). For supplementing PEPS convergence on the larger lattices, we used $D_{\max} = 750$. Although this bond dimension is not large enough to capture the ground state energy or entanglement of such large systems with high precision, we found it sufficient to capture the

first 3–4 digits of the ground state energy and to help with distinguishing between the different low-entanglement ordered phases present in the finite phase diagram, which have substantially larger gaps than the bulk system due to edge effects.

## Bulk mean field and classical phases

The mean field phase diagram for the bulk system (including all long-range interactions) in Fig. 2d was generated by the following procedure.

- Parameterize the single site wavefunction as $|\phi_i\rangle = \sin^2(\theta_i)|0\rangle + \cos^2(\theta_i)|1\rangle$, where $|0\rangle$ is the atomic ground state and $|1\rangle$ is the excited Rydberg state.
- Construct a completely un-entangled many-body wavefunction as a typical product of these single-site states according to all reasonable unit cells between size $2 \times 2$ and $8 \times 10$ (supercells are not necessary for mean-field convergence).
- Initialize all possibly relevant configurations for each unit cell as initial guesses. These can be obtained from classical algebraic arguments or classical Monte Carlo. We used a simple classical Monte Carlo Metropolis algorithm to find low energy crystals for each unit cell size.
- Minimize the Γ-point energy for all guesses with respect to the $\{\theta_i\}$ using gradient descent. Analytic gradients are easily derived, or automatic differentiation can be employed.
- Classify the phase of the lowest energy state using the same density-based order parameters as the Γ-point DMRG calculations.

The phase space was scanned with a $\delta$-resolution of 0.1 and a $R_b$-resolution of 0.025. Importantly, these calculations are subject to the same limitation as the Γ-point DMRG - they do not capture any possible low energy states with a unit cell larger than $8 \times 10$. Although such states are not expected in the phase space under examination, this study cannot definitively rule them out.

The classical phase diagram for the bulk system (including all long-range interactions) in Fig. 2c was generated by the following procedure.

- Run classical Monte Carlo minimization of the Γ-point energy for every unit cell size between $2 \times 2$ and $10 \times 10$ at phase space points spaced by $\Delta\delta = 0.3$, $\Delta R_b = 0.1$.
- For all low energy configurations obtained at all phase points, derive their continuous functional form $E(\delta, R_b)$ by numerically integrating the interactions.
- Analytically solve for the intersection line between each adjacent pair of configurations in phase space that have minimal energy.

These calculations are also subject to the same limitation as above —any states with unit cells larger than $10 \times 10$ are not captured, and we cannot rule out their possible existence.

## Data availability

Data and plotting scripts for Figs. 3, 4, and 5 can be found at https://gitlab.com/mattorourke41/rydberg_public_data. All other data is available from the authors upon request.

## Code availability

Source codes for the numerical simulations are available from the authors upon request.

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

## Acknowledgements

We thank M. Endres, J. Alicea, X. Chen, and H. Changlani for interesting discussions on Rydberg atom systems as well as entangled phases. M.J.O. acknowledges financial support from a US National Science Foundation Graduate Research Fellowship via grant DEG-1745301. G.K.C. acknowledges support from the US National Science Foundation via grant no. 2102505. G.K.C. is a Simons Investigator. Computations were conducted in the Resnick High Performance Computing Center, supported by the Resnick Sustainability Institute at the California Institute of Technology.

## Author contributions

M.J.O. and G.K.C. conceived the study. M.J.O. and G.K.C. contributed to the conceptual design of the new numerical methods. M.J.O. implemented the methods and carried out all numerical calculations. M.J.O. and G.K.C. analyzed the results and contributed to the writing of the manuscript.

## Competing interests

The authors declare no competing interests.
