## [Peer Review File · Nature Communications]

Report on “Entanglement in the quantum phases of an unfrustrated Rydberg atom array”

Recent experiments on programmable Rydberg quantum simulators have successfully probed a variety of quantum phases and phase transitions in a system of 256 qubits. In this work, O’Rourke and Chan explore the phase diagram of such Rydberg atoms arrayed on a square lattice, using a combination of variational tensor network methods, including Γ -point DMRG to study the bulk phase diagram and PEPS for large finite-size systems. They find a zoo of different symmetry-breaking ordered phases as well as an entangled “nematic” phase, and comment on comparisons to experimental data.

While the Rydberg platform itself is experimentally quite exciting, the square-lattice system has been extensively studied (see Refs. 19, 20, 49, arXiv:2112.10789), so any work on the subject at this stage can only represent incremental progress. The observation of an entangled solid phase at a large Rydberg blockade radius is perhaps unsurprising (given that Rydberg gates are anyway routinely used to prepare entangled states based on the same blockade mechanism). Ironing out the fine details of the phase diagram is a useful academic exercise for the community, but it only caters to a narrow, specialized audience. Moreover, as I note below, the reported comparisons to experimental data also do not yield any fundamentally new insights. Hence, I cannot recommend this manuscript’s publication in *Nature Communications* but would rather suggest a more specialized journal such as *Communications Physics*.

Main questions/ clarifications: In the following section, I outline a few important questions and comments that should ideally be addressed in any revised version of the manuscript (even if it is ultimately transferred to another journal in the *Nature* family).

1) Numerical methods:

Regarding Γ -point DMRG, the authors say that “*Identifying all accessible supercells which give the same ground state order... , we can ensure that all competing low-energy states are well converged w.r.t. finite size effects, and thus properly identify the true ground state.*”

A. In this method, do different phases require differently sized supercells or is the same supercell used throughout the entire phase diagram? Intuitively, if some ordered solid phase has a very large unit cell that is incompatible with the dimensions of the chosen supercell, wouldn’t this approach miss such phases?

B. In Fig. 1b, why does the energy go up as the supercell size increases from 6×8 ?

2) Phase diagram:

A. In the classical ($\Omega = 0$) phase diagram of Fig. 2(c), I find it a bit surprising that there is no “devil’s staircase” of solid phases at large detunings, Given any generic filling fraction (say, $4/19$), why can one not observe a solid with that density?

B. On line 197, it is written that “*Comparing Fig. 2b and Ref. [19], we see that having all atoms interact on an equal footing (via the Bloch basis) destroys the quantum ordered phases seen in [19].*” This is misleading—in fact, several of the phases reported in Ref. 19 (including the disordered, checkerboard, striated, and star) are also seen in Fig. 2(b).

C. For a manuscript whose central objective is to provide a detailed study of the phase diagram, it is not sufficient to just list the observed phases. The authors should also investigate the natures of the quantum phase transitions into these new phases (first-order vs. continuous, universality classes, etc.) as these details are crucial for any potential experimental realization.

3) Nematic phase and entanglement:

A. One of the key findings of the current work is the entangled nematic phase as opposed to the previously known phases that “*contain little entanglement*”. This is an interesting but perhaps not surprising result: it is already known that Rydberg atom arrays—even on the unfrustrated square lattice—can support entangled quantum ground states. For instance, in Ref. 19, it was shown that certain regions in the disordered phase can have a von Neumann entanglement entropy $\sim \mathcal{O}(1)$, which is comparable to the entanglement entropy observed for the nematic phase in Fig. 3(c).

B. The authors emphasize that the striated phase is qualitatively “*a mean-field state, confirmed by the match between the mean-field and exact correlation functions both at $(\delta, R_b) = (3.1, 1.5)$* ”. However, this statement, which is based on only a single data point, is presumably only true deep in the phase. How well does the mean-field ansatz perform closer to the phase boundary with the disordered phase? Moreover, note that recent work (Miles *et al.*, arXiv:2112.10789 [quant-ph]) has claimed that the experimentally prepared striated phase is also entangled and cannot be described by a product state, which detracts significantly from the novelty of the nematic phase.

C. Another question is whether the entanglement seen in the nematic phase is long-ranged or short-ranged. Genuinely long-range quantum entanglement—like in a quantum spin liquid—would be a novel feature, whereas short-range entanglement is relatively uninteresting. For example, I would suspect that even the various boundary-bulk frustrated ground states shown in Fig. 4(c) have nonzero short-ranged entanglement.

D. Finally, what are the ordering wavevectors of the nematic phase?

4) Experimental connections:

A. The authors write “*Our calculations support a re-interpretation of the experimental data with a significantly larger square/striated region and much smaller star phase.*” Again, this is not new—large-scale QMC simulations (Ref. 49) have already found that on finite lattices, the striated phase is greatly expanded at the cost of the shrunken star phase due to a boundary-ordering transition. Therefore, the results of Fig. 5 fail to add anything beyond what is already known.

B. It is argued that the finite nematic phase in Fig. 4(d) is the same as the “exact” bulk phase by comparing the correlators $\langle n_i n_j \rangle$. While I agree that Fig. 4(e) shows a difference from the mean-field ansatz, it does not clearly establish whether the finite nematic phase is related to the bulk phase or whether it is a dressed version of the classical 3-star state. To answer this question, one should compute the entanglement entropy of this finite-size state, in analogy to Fig. 3(c).

In summary, while the numerical analysis here is based on state-of-the-art methods, the findings of this manuscript are neither fundamentally new nor surprising, but rather quantitatively tweak the details of the phase diagram. Therefore, it is an interesting and valuable research contribution, but one that is not likely to be of broad interest to the general readership of *Nature Communications*. Hence, I can only recommend that the manuscript be transferred to a specialized journal.

Reviewer #2 (Remarks to the Author):

Dear Editor and Authors,

The present work accomplishes four goals. First, the Authors present a novel numerical tensor-network-based scheme for reliably determining the phase diagram of two-dimensional Rydberg atom arrays with long-range Van der Waals interactions. Second, this scheme is demonstrated in great detail for the square lattice, where the Authors identify and correct some significant flaws in previous numerical works, which arose due to truncating the long-range Rydberg interactions. Third, the Authors present a new conceptual finding: despite the simplicity of the square lattice geometry, the ground state can exhibit a quantum phase with significant quantum entanglement, and correspondingly this phase is entirely missed by classical and mean-field approximations (and was overlooked by previous work due to truncation effects). Finally, the Authors make a quantitative connection with data from recent experiments on the square lattice, where they can explain features which had been deemed puzzling, and moreover they propose a new order parameter that can be probed in future experiments.

The above four results make for an impressive work. To the best of my ability (and with the caveats mentioned in the below paragraphs), I find the Authors' analysis convincing and clear. It is commendable to see the painstaking level of detail and benchmarking that the Authors have displayed/performed. Beyond the above four specific achievements, more broadly I believe that this work sets a clear standard for the study of such Rydberg systems, which is welcome given that the experimental platform is experiencing a boom in its capabilities and attention. (The extensive SI then serves as a valuable point of reference.)

For these reasons, I am inclined to support publication in Nature Communications. Before I can fully endorse this, I would like the Authors to consider the following remarks:

1) In the abstract, the Authors claim that they show the new entangled phase can exist in a system without frustration. Perhaps it is a matter of semantics, but can one alternatively interpret the longer-range interactions as causing frustration? As a more mundane (or at least well-known) instance, it is known that the J_1 - J_2 (i.e. nearest and second nearest neighbor) Heisenberg model on the square lattice is frustrated. Hence I would ask the Authors: what precisely do the Authors mean when they say their model is not frustrated?

2) On the methodological front, it took me quite a while to digest the Authors' new Gamma-DMRG method. Here the explanation is perhaps unnecessarily foreign to the intended readership. (The

Authors speak of a 'Bloch basis', but there is not an obvious notion of momentum wave for spin operators.) If I understand correctly, are the Authors 'simply' performing DMRG on a torus, with the caveat that the way they evaluate their interactions is not truncated by the circumference of the torus but is instead infinitely-often going across the system? If not, can the Authors explain how it is different? If my characterization is correct, it would be useful to make that connection in the manuscript (and does it then make to sense to give it a totally new name like Gamma-DMRG?), and secondly, have the Authors searched the literature to see if other published works have performed torus-DMRG before on long-range systems (and if so, did those works simply truncate?)? I feel this connection and distinction with established DMRG methods is lacking, and its addition would aid the readability of the paper and increase the chances of the method being adapted by other groups.

3) The Authors' discussion of the novel nematic phase is interesting. I had two questions:

3a) I found the sentence "the state mainly composed of $|abab \dots\rangle$ configurations, which allow for greater itinerancy between different column states and thus energy lowering via σ_x " hard to digest. Do the Authors mean to say that we can have the virtual process $|abab\rangle \rightarrow |abcb\rangle \rightarrow |abab\rangle$, where at each stage we preserve the "no-aa" condition? (Unlike $|abca\rangle$ where e.g. the fluctuation $|abaa\rangle$ or $|abba\rangle$ violates that condition.) If this is what the Authors mean, it would be useful to spell it out (even if just in a footnote).

3b) One important aspect which is not clear to me from the Authors' discussion, is whether the entanglement of this nematic phase is robust in the thermodynamic limit. Is the nematic phase forming cat states along the circumference? If so, those would spontaneously collapse for large enough system sizes, and it would just be a finite-size artifact that the Authors find the entangled state? E.g., it seems the nematic state is stabilized by the virtual processes like $|abab\rangle \rightarrow |abcb\rangle \rightarrow |abab\rangle$ (see my previous paragraph), but that single virtual process requires $\sim L$ number of atoms to fluctuate, and thus for large enough circumference L it will not occur. Note that this question is important, since the Authors stress the non-trivial entangled nature of the ground state, which is only really interesting to the extent that it is not a finite-size cat state. (E.g., any of the other ordered phases the Authors find can also be represented as one of its cat states [by taking superpositions] which would exhibit a lot of entanglement.) I presume the Authors want to say that the entanglement in the nematic phase is not such a cat state artifact, but it seems to me that this requires some further discussion/clarification.

4) The Authors make a compelling case that previous experimental data had been misinterpreted and the Authors offer a new interpretation of existing data, which matches their numerical results well. However, the Authors do not address the issue of energy scales: if I understand correctly, the experiment prepares the state by sweeping through parameter space, and whatever rate they use will set some corresponding energy scale below which they cannot resolve. Correspondingly, we would not expect the experiment to be able to prepare any state whose energy gap is smaller than

that scale. Do the Authors have any information about the energy gap about their ground state(s), even just for an illustrative point of interest? While I agree with the Authors that the side-by-side comparison between experimental data and numerics is compelling, I think their argument would be made waterproof if the energy scales match out.

5) There have also been Monte Carlo works related to Rydberg atom arrays, in particular the general scheme laid out in arXiv:2107.00766 (not cited in the present manuscript) and arXiv:2112.10790 (Ref.49 in the present work). How would the Authors characterize the power and applicability of their method compared to those QMC methods?

Reviewer #3 (Remarks to the Author):

The authors study the Hamiltonian describing the physics of Rydberg atom arrays, using two tensor-network (TN) methods to obtain a ground-state phase diagram. This Hamiltonian is well agreed-upon and has been studied very recently by several groups using DMRG and Monte Carlo techniques, which aim to compare to recent experimental results. This work uses two TN techniques for this purpose, one being Gamma-point DMRG, which they invent in this paper, and the other being PEPS, which makes use of several recent technical advances. The authors' numerical techniques are more accurate than previous DMRG studies, and they find novel features in the phase diagram, including a new 'nematic' phase, which has more significantly entanglement than any of the other phases previously observed.

The work seems to be of a high caliber, with the numerical calculations being performed very carefully. This is important since there are many subtleties present, and the authors discuss them at length in the supplementary material. The results are also timely, in that the experimental platform under consideration is developing rapidly and producing results that need theoretical analysis. The nematic phase that is found is indeed a very interesting theoretical proposal which can be observed in Rydberg atom arrays in the near future. Additionally, they invent a new numerical method, Gamma-point DMRG, which can be used for future studies of long-range interacting models.

Despite the positive aspects, I find a couple big problems with this work, which I list here.

1. The Gamma-point DMRG method invented in this paper seems to be an uncontrolled method. In particular, the entanglement between supercells is neglected. This implies, among other things, that the environment tensors in the energy calculation are brutal approximations. This approach is likely

better than studying a single supercell with DMRG, as was done in previous works, but there is no way to tell for sure, since there is no systematic improvement that can cure this issue, making it an uncontrolled approximation. Unfortunately, there are no details on this and very few details on other aspects of the method in the paper, including the supplementary material, which is itself strange, since the method is presented here for the first time. Given this, I find it hard to have full faith in the results that the authors call the 'bulk phase diagram'.

2. Given the problems with the infinite-size method, the main punchline of the paper should come from the finite-size PEPS calculations. Here the authors find finite-size analogues of the bulk phases. However, in order to stabilize the most exciting phase, the 'nematic' one, a lot of 'massaging' needed to be done, i.e. the variational ansatz needed to be more heavily restricted. While the resulting state does indeed have spatial density correlations that look like they could potentially become the nematic ones in the thermodynamic limit, it is not entirely convincing. Therefore, putting everything together, I have to say that the evidence for the existence and stability of the nematic phase is quite thin.

Given these two issues, I cannot strongly recommend this paper be published in Nature Communications.

Aside from this, there are some more minor suggestions listed below.

1. The last big result of the paper is the different interpretation of the phase diagram measured in experiments, Figs. 5 and 15. Their main point seems to be in the interpretation of the star phase, where they claim that a different, more properly defined order parameter should be used that is much better at distinguishing the star phase from other phases. Using the old order parameter, the experiment, the previous DMRG results (or their reproduction via truncated PEPS) and the present PEPS method on different-size lattices all find roughly the same region of stability for the star phase. However, using the new order parameter, the present method with long-range interaction finds that the star phase is significantly diminished and only visible on large enough lattices, while the approximation of truncated interactions (akin to previous DMRG results) finds that the region of the star phase is largely unaltered. This indeed looks like a novel finding. The PEPS calculation including all the long-range Hamiltonian terms is almost certainly a more accurate calculation than the range 2 one. However, given that the experimental data is (likely) available upon request, I think the authors should try to obtain it and confirm their prediction. I understand that this relies on the experimental group's cooperation, but the results would be much stronger if row (a) of Figs. 5 and 15 had a fourth plot.

2. Another point is regarding the long-range interactions. It is clearly demonstrated in this paper that keeping the only the range 2 interactions is not a great approximation of the van der Waals term. However, it is not clear why a longer-range truncation will not suffice. For example, the Monte Carlo work of Ref. [49] experimented in the range of interactions to keep and found convergence in their results at range 4. It would be nice see a similar analysis here, so that using the long-range term can be avoided.

3. Finally, a minor comment. The fourth paragraph of the introduction is largely repeating a lot from the first. I suggest it be rewritten.

Responses to Referee Comments - Manuscript NCOMMS-22-06367-T

Below we have included the text of the referee's comments in black, and interspersed with these comments are our point-by-point replies in red.

Reviewer #2 (Remarks to the Author):

The present work accomplishes four goals. First, the Authors present a novel numerical tensor-network-based scheme for reliably determining the phase diagram of two-dimensional Rydberg atom arrays with long-range Van der Waals interactions. Second, this scheme is demonstrated in great detail for the square lattice, where the Authors identify and correct some significant flaws in previous numerical works, which arose due to truncating the long-range Rydberg interactions. Third, the Authors present a new conceptual finding: despite the simplicity of the square lattice geometry, the ground state can exhibit a quantum phase with significant quantum entanglement, and correspondingly this phase is entirely missed by classical and mean-field approximations (and was overlooked by previous work due to truncation effects). Finally, the Authors make a quantitative connection with data from recent experiments on the square lattice, where they can explain features which had been deemed puzzling, and moreover they propose a new order parameter that can be probed in future experiments.

The above four results make for an impressive work. To the best of my ability (and with the caveats mentioned in the below paragraphs), I find the Authors' analysis convincing and clear. It is commendable to see the painstaking level of detail and benchmarking that the Authors have displayed/performed. Beyond the above four specific achievements, more broadly I believe that this work sets a clear standard for the study of such Rydberg systems, which is welcome given that the experimental platform is experiencing a boom in its capabilities and attention. (The extensive SI then serves as a valuable point of reference.)

For these reasons, I am inclined to support publication in Nature Communications. Before I can fully endorse this, I would like the Authors to consider the following remarks:

- 1) In the abstract, the Authors claim that they show the new entangled phase can exist in a system without frustration. Perhaps it is a matter of semantics, but can one alternatively interpret the longer-range interactions as causing frustration? As a more mundane (or at least well-known) instance, it is known that the J_1 - J_2 (i.e. nearest and second nearest

neighbor) Heisenberg model on the square lattice is frustrated. Hence I would ask the Authors: what precisely do the Authors mean when they say their model is not frustrated?

The authors would first like to thank the reviewer for their kind comments on the context and merits of our manuscript. Regarding the notions of frustration, we acknowledge the reviewer's point that beyond-nearest-neighbor interactions can generally induce frustration in any lattice problem. Our choice of language such as "unfrustrated" and "absence of frustration" was intended to refer only to a lack of geometric frustration via the lattice. Many other recent works on Rydberg atom arrays use geometrically frustrated lattices to generate entangled states, and we are simply trying to distinguish our results from those. We have updated the manuscript to clarify this point.

2) On the methodological front, it took me quite a while to digest the Authors' new Gamma-DMRG method. Here the explanation is perhaps unnecessarily foreign to the intended readership. (The Authors speak of a 'Bloch basis', but there is not an obvious notion of momentum wave for spin operators.) If I understand correctly, are the Authors 'simply' performing DMRG on a torus, with the caveat that the way they evaluate their interactions is not truncated by the circumference of the torus but is instead infinitely-often going across the system? If not, can the Authors explain how it is different? If my characterization is correct, it would be useful to make that connection in the manuscript (and does it then make to sense to give it a totally new name like Gamma-DMRG?), and secondly, have the Authors searched the literature to see if other published works have performed torus-DMRG before on long-range systems (and if so, did those works simply truncate)? I feel this connection and distinction with established DMRG methods is lacking, and its addition would aid the readability of the paper and increase the chances of the method being adapted by other groups.

To address these valuable concerns we have added more discussion of the Gamma-point DMRG method in the main text, and included extensive additional discussion in the Supplementary Information. The Bloch basis is defined now in the main text. In our view, the clearest way to understand what we are doing is in fact to say that we are doing DMRG in a different basis, and the quantities in the original basis (in an infinite system) are related by a basis transformation. For example, by doing DMRG calculations in the Bloch basis, one can obtain correlation functions of observables in the infinite system (since the basis states are defined on the infinite lattice) but the correlation functions are constrained to be periodic functions, e.g. the pair correlation function satisfies $\langle n_i n_j \rangle = \langle n_i n_{j+R} \rangle$, where R is a supercell lattice vector. This perspective makes it clear that one is actually doing a calculation on an infinite system, but with some constraints on the Hilbert space. An alternative perspective is that the Hamiltonian in this

basis looks like the Hamiltonian on a torus, but where interactions wrap around infinitely often, as the reviewer points out. This is also a natural perspective, although it is perhaps less clear how other observables are treated. However, we have included this point in the discussion given in the SI.

Bloch bases are indeed commonly used in itinerant fermionic problems, such as the Hubbard model, particularly with techniques based on perturbation theory and quantum Monte Carlo. Their use with DMRG is less common, but we now cite a reference from electronic structure DMRG.

3) The Authors' discussion of the novel nematic phase is interesting. I had two questions:

3a) I found the sentence "the state mainly composed of $|abab \dots\rangle$ configurations, which allow for greater itinerancy between different column states and thus energy lowering via σ_x " hard to digest. Do the Authors mean to say that we can have the virtual process $|abab\rangle \rightarrow |abcb\rangle \rightarrow |abab\rangle$, where at each stage we preserve the "no-aa" condition? (Unlike $|abca\rangle$ where e.g. the fluctuation $|abaa\rangle$ or $|abba\rangle$ violates that condition.) If this is what the Authors mean, it would be useful to spell it out (even if just in a footnote).

We agree with the reviewer's assessment that the discussion of this point was not sufficiently clear. We have significantly expanded this discussion in the main text and especially the supplementary information in light of this question and the additional questions asked by the reviewer below.

3b) One important aspect which is not clear to me from the Authors' discussion, is whether the entanglement of this nematic phase is robust in the thermodynamic limit. Is the nematic phase forming cat states along the circumference? If so, those would spontaneously collapse for large enough system sizes, and it would just be a finite-size artifact that the Authors find the entangled state? E.g., it seems the nematic state is stabilized by the virtual processes like $|abab\rangle \rightarrow |abcb\rangle \rightarrow |abab\rangle$ (see my previous paragraph), but that single virtual process requires $\sim L$ number of atoms to fluctuate, and thus for large enough circumference L it will not occur. Note that this question is important, since the Authors stress the non-trivial entangled nature of the ground state, which is only really interesting to the extent that it is not a finite-size cat state. (E.g., any of the other ordered phases the Authors find can also be represented as one of its cat states [by taking superpositions] which would exhibit a lot of entanglement.) I presume the Authors want to say that the entanglement in the nematic phase is not such a cat state artifact, but it seems to me that this requires some further discussion/clarification.

We are very grateful to the reviewer for pointing out this thought-provoking point and the need for its clear discussion in the manuscript. We have spent several months pursuing various additional numerical simulations and analytical techniques to better understand the quantum nematic order in the TDL. We have made a number of changes to the manuscript to address this question. Before describing them, we first address the question of the physics.

Entanglement plays two roles in the nematic phase (which were not clearly distinguished in the original manuscript). The first is something we have now termed “entanglement stabilization”. Namely, if one goes beyond the unentangled mean-field state, using a perturbative treatment of σ_x , the excitations in the Rydberg atom lattice become itinerant. Then, starting from different crystal configurations, the 4th order correction to the energy clearly stabilizes certain configurations such as $|abab\dots\rangle$ versus the classical crystal ground state $|abcabc\rangle$. Thus, the configurations that appear in the nematic phase are all “entanglement stabilized” (relative to the configurations in the classical crystal or mean-field state); these are the configurations that are highly populated in the actual DMRG ground-state. Because of this entanglement selection, one cannot view the nematic phase as a fluctuation around the classical ground-state crystal (or mean-field state) - in fact the classical ground-state crystal configurations are almost completely absent in the nematic phase.

The second question is about the fate of “global” entanglement between the macroscopically different entanglement stabilized crystal orders. There are multiple possibilities in the TDL, ranging from a cat state, to an exotic quantum liquid, to something in between, e.g. with algebraically decaying correlations. We have confirmed that the nematic state we report on the finite 12×9 supercell is gapped using exact diagonalization techniques, ruling out a cat state for that supercell size (although not necessarily in the TDL). We note that DMRG disfavors cat states in general because it is a low-entanglement ansatz, and we see no cat states in any of the ordered phases. Despite this bias against cat states, the DMRG calculations produce significant entanglement in the nematic phase and the configurational analysis shows that not only do all the permutations of the $|abab\dots\rangle$ configurations appear in the ground-state with equal weight, but there are large weights for configurations like $|acab\dots\rangle$ and with other column defects. The entanglement spectrum is also consistent with a model state (see SI) with interesting features. All this indicates that there are quantum fluctuations and entanglement in this state not simply of cat character; and the large energetic contribution due to itinerancy is consistent with the possibility of some form of melting, as seen in the floating incommensurate solid phase of 1D Rydberg atom chains. However, given the finite size of all our calculations, we also cannot definitively rule out that when coarse grained over sufficiently long distances and in the TDL, the nematic phase is just a strong fluctuation around an entanglement stabilized crystalline state.

We have extensively updated the main text and SI to include more details and explanations of our observations and new insights. Primarily, in the main text we now carefully distinguish the two different entanglement scenarios above, and refrain from making definitive claims about the TDL stability of a “globally” entangled state. We have also added multiple sections to the SI describing our perturbative analysis and a simple globally entangled model state that has similar entanglement features to our computed DMRG nematic state.

4) The Authors make a compelling case that previous experimental data had been misinterpreted and the Authors offer a new interpretation of existing data, which matches their numerical results well. However, the Authors do not address the issue of energy scales: if I understand correctly, the experiment prepares the state by sweeping through parameter space, and whatever rate they use will set some corresponding energy scale below which they cannot resolve. Correspondingly, we would not expect the experiment to be able to prepare any state whose energy gap is smaller than that scale. Do the Authors have any information about the energy gap about their ground state(s), even just for an illustrative point of interest? While I agree with the Authors that the side-by-side comparison between experimental data and numerics is compelling, I think their argument would be made waterproof if the energy scales match out.

In the experiment, the phase diagram that we compare to is prepared by sweeping $\bar{\delta}$ over a range of 30 MHz in a time of $T = 2.5$ microseconds. In units of their experimental value of Ω , this can be written as $T \approx 62.8 / \Omega$. From our numerical results on the same 13×13 lattice as the experiment, we can compute the energy gap between the square order and star order at a phase point $(\bar{\delta}, R_b) = (3.7, 1.7)$, where the experimental paper identifies the star order but we find the square order (albeit close to the boundary). In units of Ω , we find that the total energy of the square order is lower by $\approx 3.27 \times \Omega$. Based on Heisenberg energy-time uncertainty, we roughly expect the energy resolution of the experimental sweep to be $\sim 1/T = \Omega / 62.8$. From this simple analysis, we see that the energy gap between the square and star orders is far greater than the energy scale that cannot be resolved experimentally. This lends additional support to the notion that the experimental data is misinterpreted, and they do not in fact prepare the star order over the whole domain that they label.

Of course, the above analysis is extremely simple and does not include in any detail many of the complicated choices made in the experiment, such as the use of splined sweep functions whose form is “optimized based on maximizing the respective order parameter”. Given the difficulty of

including such effects numerically and the rough nature of the above analysis, we have decided to forgo the inclusion of this point in the manuscript.

5) There have also been Monte Carlo works related to Rydberg atom arrays, in particular the general scheme laid out in arXiv:2107.00766 (not cited in the present manuscript) and arXiv:2112.10790 (Ref.49 in the present work). How would the Authors characterize the power and applicability of their method compared to those QMC methods?

We thank the reviewer for pointing out the citation that we missed. We have now also included additional references to other Monte Carlo and machine-learning based techniques that have very recently been applied to studying the phases of the 2D Rydberg atom system, e.g. [Czischek, et. al. Phys Rev B, 105 205108 (2022)] and [Miles, et. al. arXiv:2112.10789 (2021)].

The observation that the special form of the Rydberg Hamiltonian allows for sign-free MC simulation is an important advance. These will no doubt be important tools for ground-state and equilibrium simulations. Nonetheless, there are other subtleties to efficient MC simulation that are already noted in the above articles. In other areas, for example in the Hubbard model, the subtleties of sign-free QMC at large U (e.g. large variance) are well known.

In general terms, the methods we have presented are not limited to sign-free Hamiltonians and can be applied to dynamics. These are the general strengths of tensor networks relative to QMC, and of course dynamics are of interest in Rydberg arrays. The performance of sign-free QMC when there is an emergent low energy Hamiltonian which may have a sign problem is also an interesting question.

Reviewer #3 (Remarks to the Author):

The authors study the Hamiltonian describing the physics of Rydberg atom arrays, using two tensor-network (TN) methods to obtain a ground-state phase diagram. This Hamiltonian is well agreed-upon and has been studied very recently by several groups using DMRG and Monte Carlo techniques, which aim to compare to recent experimental results. This work uses two TN techniques for this purpose, one being Gamma-point DMRG, which they invent in this paper, and the other being PEPS, which makes use of several recent technical advances. The authors' numerical techniques are more accurate than previous DMRG studies, and they find novel features in the phase diagram, including a new 'nematic' phase, which has more significantly entanglement than any of the other phases previously observed.

The work seems to be of a high caliber, with the numerical calculations being performed very carefully. This is important since there are many subtleties present, and the authors discuss them at length in the supplementary material. The results are also timely, in that the experimental platform under consideration is developing rapidly and producing results that need theoretical analysis. The nematic phase that is found is indeed a very interesting theoretical proposal which can be observed in Rydberg atom arrays in the near future. Additionally, they invent a new numerical method, Gamma-point DMRG, which can be used for future studies of long-range interacting models.

The authors would like to thank the reviewer for their kind comments on the context and merits of our manuscript.

Despite the positive aspects, I find a couple big problems with this work, which I list here.

1. The Gamma-point DMRG method invented in this paper seems to be an uncontrolled method. In particular, the entanglement between supercells is neglected. This implies, among other things, that the environment tensors in the energy calculation are brutal approximations. This approach is likely better than studying a single supercell with DMRG, as was done in previous works, but there is no way to tell for sure, since there is no systematic improvement that can cure this issue, making it an uncontrolled approximation. Unfortunately, there are no details on this and very few details on other aspects of the method in the paper, including the supplementary material, which is itself strange, since the method is presented here for the first time. Given this, I find it hard to have full faith in the results that the authors call the 'bulk phase diagram'.

To address the valuable concerns expressed in point 1, we have added more discussion of the Gamma-point DMRG method in the main text, and included extensive additional discussion in the Supplementary Information.

The key points are:

(i) The reviewer states that the entanglement between supercells is neglected. As we now discuss in the text, by choosing the Bloch basis instead of the typical localized site-basis used in tensor network calculations, our simulations intrinsically include entanglement between different supercells without needing to generate “bulk environment” tensors like typical iPEPS or iDMRG methods. This entanglement is of a constrained form (as it is in an iPEPS with finite bond dimension) but can be systematically improved by increasing the supercell size. In general, the correlations between supercells induced by the Bloch basis are too strong, rather than too weak as in an iPEPS with small bond dimension.

(ii) We are not sure why the reviewer states there is no systematic improvement that can cure the issue. Overall, the methods are systematically converged by increasing the supercell size and MPS bond dimension. This is similar to the standard use of DMRG to study bulk physics, see e.g. [Science 358, Issue 6367 pp. 1155 (2017)]. Increasing the supercell size in the Bloch basis is exactly the same as converging Brillouin zone (“k-pt”) sampling, for example, as is used in all electronic structure calculations of materials (see e.g. the k-pt sampling plot in Fig. 1F in [Science 377, 1192 (2012)]). What is happening when increasing the number of kpts/supercell size is that the description of the infinite bulk problem becomes less and less constrained (similar to increasing the bond dimension in an iPEPS) and eventually the unconstrained bulk result is obtained. The important point for convergence is that the supercell size is sufficiently large compared to the size of the order observed.

The main difference between the calculations done in the Bloch basis and earlier work done in finite cluster DMRG, is that by working in the infinite system at the outset, we are not conflating the finite size error due to the wrong Hamiltonian (as used in finite cluster DMRG due to truncation of interactions) and the error due to lack of convergence of the wavefunction. Because the first error is larger than the second, and indeed as the numerics show, this leads to much more rapid convergence to the TDL. We believe that our extensive new discussion, along with our careful demonstration of our systematic convergence strategy in Supplementary Information Fig. 6 should give confidence in the accuracy of our bulk phase diagram results.

2. Given the problems with the infinite-size method, the main punchline of the paper should come from the finite-size PEPS calculations. Here the authors find finite-size analogues of the bulk phases. However, in order to stabilize the most exciting phase, the ‘nematic’ one, a lot of ‘massaging’ needed to be done, i.e. the variational ansatz needed to be more heavily

restricted. While the resulting state does indeed have spatial density correlations that look like they could potentially become the nematic ones in the thermodynamic limit, it is not entirely convincing. Therefore, putting everything together, I have to say that the evidence for the existence and stability of the nematic phase is quite thin.

We agree that correspondence between the finite-size “nematic” state and the corresponding bulk state is not entirely convincing. This is why we use the terminology “the correlation functions reveal qualitative similarities to the bulk nematic phase”. We are not making any claim stronger than that statement for the finite state that we manage to stabilize.

We hope that the reviewer no longer thinks the “main punchline” of the paper is solely focused on this finite-size state, due to our new, expanded discussion of the gamma-point DMRG method and bulk nematic phase stability in the text, SI, and this response letter to all reviewers. We think there are multiple other “punchlines”, such as the bulk nematic phase, direct comparisons to experiment, and the first demonstration of ground state PEPS calculations for a long-range interacting Hamiltonian.

Given these two issues, I cannot strongly recommend this paper be published in Nature Communications.

Aside from this, there are some more minor suggestions listed below.

1. The last big result of the paper is the different interpretation of the phase diagram measured in experiments, Figs. 5 and 15. Their main point seems to be in the interpretation of the star phase, where they claim that a different, more properly defined order parameter should be used that is much better at distinguishing the star phase from other phases. Using the old order parameter, the experiment, the previous DMRG results (or their reproduction via truncated PEPS) and the present PEPS method on different-size lattices all find roughly the same region of stability for the star phase. However, using the new order parameter, the present method with long-range interaction finds that the star phase is significantly diminished and only visible on large enough lattices, while the approximation of truncated interactions (akin to previous DMRG results) finds that the region of the star phase is largely unaltered. This indeed looks like a novel finding. The PEPS calculation including all the long-range Hamiltonian terms is almost certainly a more accurate calculation than the range 2 one. However, given that the experimental data is (likely) available upon request, I think the authors should try to obtain it and confirm their prediction. I understand that this relies on the experimental group’s cooperation, but the results would be much stronger if row (a) of Figs. 5 and 15 had a fourth plot.

Unfortunately, we have reached out to the experimental group multiple times, but have not been able to obtain their data. We agree that the suggested plot would be very nice to have.

2. Another point is regarding the long-range interactions. It is clearly demonstrated in this paper that keeping the only the range 2 interactions is not a great approximation of the van der Waals term. However, it is not clear why a longer-range truncation will not suffice. For example, the Monte Carlo work of Ref. [49] experimented in the range of interactions to keep and found convergence in their results at range 4. It would be nice see a similar analysis here, so that using the long-range term can be avoided.

We think that the referenced work is an impressive paper, and we too appreciate their study of the long-range effects. Of course, their conclusion that range 4 is sufficient depends on the fact that they only chose to study the checkerboard, striated, and star phases. If they had explored larger values of R_b , as we did, they almost certainly would have needed to keep longer interaction terms. We prefer to just keep all the terms, so that we do not have to perform new detailed analyses whenever we want to increase R_b , like the referenced paper. Importantly, Fig. 3 shows that the physics in the region of the nematic phase is sensitive to energy scales at least on the order of 10^{-4} per site. At an R_b value of 2.3, this corresponds to keeping interactions up to at least range ~ 11 .

3. Finally, a minor comment. The fourth paragraph of the introduction is largely repeating a lot from the first. I suggest it be rewritten.

This is true. The reason for this is that our understanding of the Nature article style is that the first paragraph (summary paragraph) should function as a standalone summary, including both the main context and main findings. Thus, our subsequent paragraphs in the introduction are expanding on the first paragraph, with unfortunately a little repetition for clarity. To reduce the repetition, we have removed a little bit from the first paragraph in our edited text.

Reviewer #1 (Remarks to the Author):

Recent experiments on programmable Rydberg quantum simulators have successfully probed a variety of quantum phases and phase transitions in a system of 256 qubits. In this work, O'Rourke and Chan explore the phase diagram of such Rydberg atoms arrayed on a square lattice, using a combination of variational tensor network methods, including Γ -point DMRG to study the bulk phase diagram and PEPS for large finite-size systems. They find a zoo of different symmetry-breaking ordered phases as well as an entangled "nematic" phase, and comment on comparisons to experimental data.

While the Rydberg platform itself is experimentally quite exciting, the square-lattice system has been extensively studied (see Refs. 19, 20, 49, arXiv:2112.10789), so any work on the subject at this stage can only represent incremental progress. The observation of an entangled solid phase at a large Rydberg blockade radius is perhaps unsurprising (given that Rydberg gates are anyway routinely used to prepare entangled states based on the same blockade mechanism). Ironing out the fine details of the phase diagram is a useful academic exercise for the community, but it only caters to a narrow, specialized audience. Moreover, as I note below, the reported comparisons to experimental data also do not yield any fundamentally new insights. Hence, I cannot recommend this manuscript's publication in Nature Communications but would rather suggest a more specialized journal such as Communications Physics.

While we respect the views of the reviewer, we would like to distinguish between two aspects of the significance of our work. The first is that from the viewpoint of numerics, it is (to our knowledge) the first demonstration of large-scale 2D tensor network calculations with long range interactions. The second is from the viewpoint of Rydberg atom physics, and this is where the reviewer is concerned about significance. In this setting, the question of significance depends on whether it is important to get the "right" result for a problem, or the "first" result.

Our work is about getting the right result. In the case of the 2D square lattice Hubbard model, for example, it seems clear that everyone agrees that it is important to get the right result about something rather than the first result. For the 2D square lattice Rydberg atom array, perhaps there is not universal consensus on the importance of getting the right result. But at the very least, because of the methodological significance of our work, it illustrates **how** one might get the right result.

Main questions/ clarifications: In the following section, I outline a few important questions and comments that should ideally be addressed in any revised version of the manuscript (even if it is ultimately transferred to another journal in the Nature family).

1) Numerical methods:

Regarding Γ -point DMRG, the authors say that *"Identifying all accessible supercells which give the same ground state order. . . , we can ensure that all competing low-energy states are well converged w.r.t. finite size effects, and thus properly identify the true ground state."*

A. In this method, do different phases require differently sized supercells or is the same supercell used throughout the entire phase diagram? Intuitively, if some ordered solid phase has a very large unit cell that is incompatible with the dimensions of the chosen supercell, wouldn't this approach miss such phases?

The supercells are large enough to contain multiple copies of the orders studied. Indeed if there is some truly long wavelength order that cannot fit into the largest supercell, then it would be missed: this is a characteristic of all numerical calculations on competing orders, see e.g. [Science 358, Issue 6367 pp. 1155 (2017)]. However, the emergence of such a competing state in the parameter regime we have studied is unexpected for multiple reasons. Primarily, extensive work by others on the one-dimensional Rydberg system for the past 10+ years, in addition to famous works on related 1D and 2D classical systems by Per Bak et. al., suggest that short-wavelength orders with rational fraction densities ($1/n$) heavily dominate the low-energy physics. In the parameter regimes we study, long-wavelength orders that we would miss are expected to be on energy scales smaller than that associated with the largest supercell, in our case on the order of 10^{-5} .

B. In Fig. 1b, why does the energy go up as the supercell size increases from 6×8 ?

Increasing the Γ point supercell size is equivalent to increasing Brillouin zone sampling in a reciprocal-space calculation. We have attempted to make this (and related) points more clear by including additional discussion of our methods in the main text and an extensive new section in the Supplementary Information. The error of Brillouin zone sampling can be related to the error of approximating the Brillouin zone integrals by quadrature; it is then well known that the

energy does not change variationally during this process; it may converge from above or below to the TDL.

2) Phase diagram:

A. In the classical ($\Omega = 0$) phase diagram of Fig. 2(c), I find it a bit surprising that there is no “devil’s staircase” of solid phases at large detunings, Given any generic filling fraction (say, $4/19$), why can one not observe a solid with that density?

This point is rather subtle. In 1D, a complete “devil’s staircase” of ground states exists in the classical limit, as famously proved by Per Bak et. al. However, even in 1D the phase space domains of stability for filling fractions p/q are well-known to be extremely small for $p > 1$ compared to $p=1$. See e.g. [Bak et. al. Phys Rev Lett 49, 4 (1982)] and [Weimer et. al. Phys Rev Lett 105, 230403 (2010)]. In 2D on the square lattice, the situation becomes much more complicated because the geometry of the lattice is very unfavorable for certain filling fractions (note the small or nonexistent domains even for some rational fractions like $1/3$ and $1/7$ in our work). It is possible that we miss very tiny slivers of different p/q in our plotted classical phase diagram (although we do detect $2/9$), but it is also likely that the square lattice completely destabilizes many filling-fractions within the parameter regime we examine. Note that this is an important difference between 1D and 2D that gives rise to 2D phases like the striated phase that have no 1D analog.

B. On line 197, it is written that “Comparing Fig. 2b and Ref. [19], we see that having all atoms interact on an equal footing (via the Bloch basis) destroys the quantum ordered phases seen in [19].” This is misleading—in fact, several of the phases reported in Ref. 19 (including the disordered, checkerboard, striated, and star) are also seen in Fig. 2(b).

This sentence is meant to be read in conjunction with the previous sentence which ends “at larger R_b ”. To clarify the confusion, we have added “at larger R_b ” to the end of the current sentence, i.e. some phases seen in [19] at larger R_b .

C. For a manuscript whose central objective is to provide a detailed study of the phase diagram, it is not sufficient to just list the observed phases. The authors should also investigate the natures of the quantum phase transitions into these new phases (first-order vs. continuous, universality classes, etc.) as these details are crucial for any potential experimental realization.

While it would be ideal to characterize all the phase boundaries, the amount of numerical work to do so accurately, and for all the phase boundaries (approximately 10) is unrealistic for this paper. In our experience, it is common even in some of the most definitive studies on

competing phases to separate the description of order of phase transitions to a separate paper (see e.g. [Science 358, Issue 6367 pp. 1155 (2017)] which is only about establishing the order of a single phase!). Additionally, even in 1D many of the phase transitions in the Rydberg chain are known to be very detailed and complicated, and are still actively being researched (see e.g. [Nature Comm. 12, Article 414 (2021)]). Proper studies of this level for the 2D lattice require additional independent studies. Nonetheless, we have provided a limited examination of some phase transitions in the SI.

3) Nematic phase and entanglement:

A. One of the key findings of the current work is the entangled nematic phase as opposed to the previously known phases that “*contain little entanglement*”. This is an interesting but perhaps not surprising result: it is already known that Rydberg atom arrays—even on the unfrustrated square lattice—can support entangled quantum ground states. For instance, in Ref. 19, it was shown that certain regions in the disordered phase can have a von Neumann entanglement entropy $\sim O(1)$, which is comparable to the entanglement entropy observed for the nematic phase in Fig. 3(c).

We have greatly expanded our discussion of the nematic order in the main text and SI, as described in detail in our responses to other reviewers’ comments. Indeed, the disordered phase has some entanglement due to itinerancy, as already pointed out in the paper. The question of interest is whether there are any interesting effects of entanglement in the ordered phases.

As now explained more clearly, the nematic phase is the only ordered phase in the phase diagram which requires a treatment of entanglement. To use a more evocative terminology, it is an entanglement stabilized phase. Microscopically, itinerancy stabilizes and selects certain configurations of Rydberg atoms, while disfavoring the classical and mean-field crystal configurations in that part of the phase diagram; these configurations then go on to populate the DMRG ground-state.

An additional feature of the nematic phase is that there are, in general, substantially larger quantum fluctuations than in any of the neighbouring ordered phases. At least at the system sizes studied, only this phase, out of all the phases studied, shows substantial global fluctuations (i.e. there is a large mixing over an exponential manifold of states). The entanglement spectrum is also consistent with a model state with interesting entanglement now described in the SI. Whether or not this coarse-grains out in the thermodynamic limit to fluctuations around an entanglement-stabilized crystal or something more exotic cannot be

conclusively determined at present. This discussion is now expanded on in both the main text and the SI.

B. The authors emphasize that the striated phase is qualitatively “a mean-field state, confirmed by the match between the mean-field and exact correlation functions both at $(\delta, Rb) = (3.1, 1.5)$ ”. However, this statement, which is based on only a single data point, is presumably only true deep in the phase. How well does the mean-field ansatz perform closer to the phase boundary with the disordered phase? Moreover, note that recent work (Miles et al., arXiv:2112.10789 [quant-ph]) has claimed that the experimentally prepared striated phase is also entangled and cannot be described by a product state, which detracts significantly from the novelty of the nematic phase.

The first question is regarding the validity of the mean-field ansatz for the striated phase within our own results. As the reviewer states themselves, our emphasis is that the physics of the striated state, along with most other ordered states, is *qualitatively* mean-field in nature. We believe this is compellingly demonstrated in Fig. 2 in the main text. Although the mean-field phase boundaries between different ordered states are not exactly the same as the DMRG phase boundaries, their correspondence is very clear and qualitatively correct. In other words, the stability of the striated phase over all other competing ordered phases can be predicted from a purely mean-field picture. As we already discuss in the main text, the disordered phase cannot be described by the mean-field ansatz, and thus neither can the order-disorder phase transition. So, as we tune $\bar{\delta}$ very close to the order-disorder transition we expect the mean-field ansatz to break down. Indeed we observe that the correlation functions get slightly worse at point $(\bar{\delta}, Rb) = (2.2, 1.5)$. We want to emphasize that this is not only true for the striated phase but also all other ordered phases that appear in the mean-field phase diagram. This can be seen clearly in the second figure of the SI (Fig. 7), where the checkerboard, striated, star, and staggered phases are all shown to have the same characteristic increase in entanglement as $\bar{\delta}$ is tuned close to the order-disorder transition. We do not believe that the mean-field nature of the ordered striated state (or any other ordered state) is invalid or misleading because the mean-field ansatz cannot properly describe the order-disorder phase transition. As a final point on this matter, we note that this is all very different from the nematic state, where the mean-field ansatz stabilizes a qualitatively different order than the entangled DMRG calculations.

The reviewer also points out the work of Miles et. al. In that work the authors do indeed make a claim that describing the striated phase “calls for a description beyond a simple product state ansatz.” This claim is made from analyzing the real experimental data from a finite Rydberg array. We note that the authors qualify their claim with statements like “[entanglement can] be generated in the dynamical state preparation process due to the quantum Kibble-Zurek

mechanism, where nonadiabatic processes can coherently generate superpositions including excited states that generically result in entanglement.” They also perform 2D DMRG calculations on a 9x9 lattice (open boundaries) to compare, but their results show that there is only significant entanglement near phase transition boundaries, and not within the ordered region that they identify as the striated phase. Indeed, they write “[entanglement] might be present in the ground state itself, particularly in the vicinity of a second-order quantum phase transition.” We therefore do not think that their results are in any disagreement with our understanding of the mean-field ansatz, as it is explained above. We would like to again mention that this does not detract from the novelty of the nematic phase, since it has qualitatively important entanglement features that persist deep into the ordered phase.

C. Another question is whether the entanglement seen in the nematic phase is long-ranged or shortranged. Genuinely long-range quantum entanglement—like in a quantum spin liquid—would be a novel feature, whereas short-range entanglement is relatively uninteresting. For example, I would suspect that even the various boundary-bulk frustrated ground states shown in Fig. 4(c) have nonzero short-ranged entanglement.

As pointed out above, entanglement is doing something interesting in the nematic phase, it is stabilizing certain configurations over the classical crystal and mean-field solution, and such entanglement stabilization is, in our view, already a novel feature. Whether or not there are additional long-range entanglement features is difficult to say; they appear to be present at the system size studied, and the entanglement spectrum has interesting properties (see SI) but a detailed finite size scaling analysis will be required in the future. Nonetheless, this is a promising place in the phase diagram to look for interesting entanglement features.

D. Finally, what are the ordering wavevectors of the nematic phase?

The primitive wavevectors for the dominant $|ababab\dots\rangle$ -type configurations in the nematic state are given by: $(4x, 3y)$. This can be used to distinguish the nematic order from the competing classical and mean-field order, which has primitive wavevectors: $(6x, 3y)$.

4) Experimental connections:

A. The authors write “*Our calculations support a re-interpretation of the experimental data with a significantly larger square/striated region and much smaller star phase.*” Again, this is not new—large-scale QMC simulations (Ref. 49) have already found that on finite lattices, the striated phase is greatly expanded at the cost of the shrunken star phase due to a

boundary-ordering transition. Therefore, the results of Fig. 5 fail to add anything beyond what is already known.

While the referenced work is an impressive paper, it is in fact primarily focused on characterizing the nature of the order-disorder transitions for the checkerboard, striated, and star phases by using finite arrays with *periodic* boundary conditions. It also contains a small section on the effects of open boundaries, which itself primarily focuses on the fact that the boundary orders before the bulk (as δ is swept from negative to positive). This is an important point that we have now cited in our own work. While the reviewer claims that the referenced work shows the striated phase is “greatly expanded”, the data in the paper (Fig. 5) only clearly demonstrates a growth of the striated phase in the δ -direction by ~ 0.2 compared to the periodic boundary calculations. The only data in the paper showing the extent of the star phase is Fig. 1, which in fact shows a large domain of stability for the star phase relative to our paper. We disagree with the reviewer’s notion that our results fail to add anything beyond what is already known. Our work 1) reveals that a large region of the “striated” domain actually corresponds to a ground state that is not striated at all (the “square order”), and is not stable on bulk or periodic lattices; 2) demonstrates the inadequacy of the Fourier-based order parameters on finite lattices (used in the original experiment as well as the referenced work in this comment) for precisely distinguishing the striated, square, and star orders; 3) shows that the true extent of the star phase on the 13x13 experimental lattice is exceptionally small due to its competition with the square order.

B. It is argued that the finite nematic phase in Fig. 4(d) is the same as the “exact” bulk phase by comparing the correlators $\langle n_{ij} \rangle$. While I agree that Fig. 4(e) shows a difference from the meanfield ansatz, it does not clearly establish whether the finite nematic phase is related to the bulk phase or whether it is a dressed version of the classical 3-star state. To answer this question, one should compute the entanglement entropy of this finite-size state, in analogy to Fig. 3(c).

We appreciate this concern, and agree that correspondence between the finite-size state and the bulk state is not entirely convincing. This is why we use the terminology “the correlation functions reveal qualitative similarities to the bulk nematic phase”. We are not making any claim stronger than that statement. Computing the bi-partite entanglement entropy from PEPS wavefunctions in a similar manner to Fig. 3, as the reviewer advises, is unfortunately computationally infeasible. That is why we have only displayed the correlation functions.

In summary, while the numerical analysis here is based on state-of-the-art methods, the findings of this manuscript are neither fundamentally new nor surprising, but rather quantitatively tweak the details of the phase diagram. Therefore, it is an interesting and

valuable research contribution, but one that is not likely to be of broad interest to the general readership of Nature Communications. Hence, I can only recommend that the manuscript be transferred to a specialized journal.

REVIEWER COMMENTS

Reviewer #1 (Remarks to the Author):

Firstly, I appreciate the authors taking the time to respond to my comments and clarify various points in the manuscript. These changes have certainly improved the presentation of the paper. However, I stand by my assessment that the work is better suited for a more specialized journal than Nature Communications and continue to recommend against publication.

I understand that this is a subjective opinion, and I respect the authors' right to maintain a different viewpoint, but let me mention four key reasons that inform my decision:

1. As the authors themselves write, "However, given the finite size of all our calculations, we also cannot definitively rule out that when coarse grained over sufficiently long distances and in the TDL, the nematic phase is just a strong fluctuation around an entanglement stabilized crystalline state." If the authors cannot establish that their "discovered" phase is truly a stable phase of matter in the thermodynamic limit, that dilutes one of the key results of this manuscript.
2. A generic quantum state is likely entangled to some degree. Just the presence of entanglement alone is not sufficient to make a state novel. The authors concede that "Whether or not there are additional long-range entanglement features is difficult to say": satisfactorily answering this question is a prerequisite to demonstrate that the nematic state is an interesting phase of matter. If, as above, it is just a finite-size manifestation of "an entanglement stabilized crystalline state", that would also appear entangled in numerical simulations despite being a rather uninteresting state.
3. The authors seem to agree in their response that the bulk nematic and the finite-size phase are not the same. This point is almost glossed over and rather subtly underemphasized in the sentence "the correlation functions reveal qualitative similarities to the bulk nematic phase". If the finite-size phase is not even the true bulk entangled phase, then I fail to see why the properties of the idealized bulk phase are even relevant to experiments on finite systems.
4. Beyond the observation of the nematic phase, all of the other results in the paper—such as the reinterpretation of the experimental results or the slight modification of phase boundaries—are, in my opinion, incremental improvements. I respect the numerical efforts that the authors have invested in Γ -point DMRG, which might be a useful technique. However, I do not believe that

the development of a numerical method alone merits publication in Nature Communications unless accompanied by significant new results that were inaccessible previously.

Reviewer #2 (Remarks to the Author):

I thank the referees for addressing my points. The manuscript and the SI have been improved. I also still stand by the positive evaluation I gave in my first work: this numerical work sets an excellent standard for the booming field of many-body physics in Rydberg simulators, and this improved method is evidenced by the authors discovering a phase of matter which had been overlooked in previous work, and which is stabilized by a different mechanism than the previously-identified ordered phases.

However, I strongly feel there are still two important issues that need to be resolved before I can recommend publication.

1) The first issue concerns the nematic phase. I appreciate the authors' extended discussion, both in their reply to my report, as well as the updated manuscript. However, I think certain aspects have not been accurately characterized:

First, the authors point out that the nematic states are not selected at the classical level but instead by a 4th order process in perturbation theory, and the authors calls this an 'entanglement-stabilized phase'. I think I agree with this physical process, but it is dangerous/ill-advised to give a new name to a well-established process: I believe this is an instance of 'order from disorder'. In the quantum context, order from disorder works by starting with an extensively degenerate space of states (usually resulting from some classical interactions), and this degeneracy is then lifted by some additional quantum fluctuations (due to some classical/ordered states allowing for more fluctuations than other states). (E.g., see the discussion for the hexagonal lattice FFIM in [Moessner, Sondhi, Chandra, PRL 84, 4457 (2000)].) This general process seems to coincide with the mechanism found by the authors.

Second, the authors say they are not sure whether the macroscopic entanglement they find in the nematic phase is a physically (un)stable (related to my question in my previous report). However, given the authors' extended discussion of this phase, it is now rather apparent to me that it is indeed unstable cat state entanglement, which can thus not extend to the thermodynamic limit. A

characteristic property of cat state entanglement is that measuring, say, a single site, will collapse the global entanglement. This is clearly the case for the nematic state found by the authors. (E.g., a single-site measurement can reveal that a whole column is in the 'a' state.) Similarly, note that the "simple product state" the authors consider in the SI below Eq.8 is in fact a cat state along the vertical direction. If these points do not already convince the authors of the unphysical nature of their entangled state, let me add two more equivalent characterizations: if one were to re-do the numerical simulations with an offset chemical potential added to only *a single site* (which favors an occupied site), the ground state will no longer have the observed entanglement; equivalently, one can find that the entangled state found by the authors has long-range order in a two-point function which does not satisfy the clustering/factorization property. It would be very unfortunate to publish a paper which can cause confusion (in the community) about the entangled nature of a state when the accurate characterization of its entanglement is within reach.

In conclusion, I believe the authors have discovered an ordered phase which is stabilized by 'order from disorder' in a Rydberg atom array. This phase is not characterized by a significant amount of entanglement, but it is still interesting as an instance of 'order from disorder'.

2) While I do not doubt the correctness of the authors' Gamma-DMRG method, its discussion is sub-par.

First, I do not yet see why the authors' definition of the Bloch basis is well-defined. The authors are considering an infinite tensor product Hilbert space $H = \otimes H_n$, where H_n is a two-level qubit. The authors define the 'Bloch' states of the form $|\tilde{0}\rangle = |0\rangle + |L\rangle + |2L\rangle + \dots$. Note that these states are not defined in the many-body Hilbert space, nor are products of such states elements in the many-body Hilbert space. (Note that in contrast, in fermionic systems one can define Fourier operators/modes $c_k = c_x + c_{x+L} + \dots$ et cetera, and then define many-body states as $c_k |0\rangle$ where $|0\rangle$ is the many-body vacuum, which is a valid state in the many-body Hilbert space. Hardcore bosons do not allow for such a treatment.) The only way I (personally) can make sense of the authors definitions is by interpreting everything in a finite-dimensional Hilbert space of a $L \times L$ torus with periodic boundary conditions. If the authors do not wish to use such a torus geometry/Hilbert space, they have to include/discuss a more careful definition of their Hilbert space.

Second, in their reply to my report, the authors also agreed that the torus perspective is "also a natural perspective, although it is perhaps less clear how other observables are treated. However, we have included this point in the discussion given in the SI." I cannot find a discussion of it in the SI (other than a discussion about periodicity of correlation functions). Also, it is not clear to me what the authors mean by "it is perhaps less clear how other observables are treated". Let me then rephrase my original question: if one were to perform torus DMRG (with interactions wrapping around rather than being truncated), would the simulations/code/results exactly coincide with what

the authors have done? If yes: then why not state that more explicitly, and why give a new name to torus DMRG? If no: can the authors pinpoint a difference?

Let me state that if the authors' method coincided with torus DMRG, I do not think that hurts the present work. Clearly the authors have used this tool in a novel way and shown it to be superior to previous treatments. But I do think it would be bad form to give a new name to an existing method, as it would very likely cause confusion in the literature. I thus encourage the authors to either clearly acknowledge and state that their method is torus DMRG, or pinpoint why it is not.

Reviewer #3 (Remarks to the Author):

The authors have replied to all my comments in a satisfactory fashion.

I just have one more comment regarding the errors generated by the Gamma-point DMRG method. It is true, and now explained clearly in the SI, that there are two systematic sources of error in the method: supercell size and bond dimension. The authors have clearly explained that convergence in both is required, and have demonstrated such convergence numerically, thereby solidifying their infinite system phase diagram. However, I want to note that because of the constrained form of the entanglement allowed between supercells, the amount of entanglement neglected for a fixed supercell size is not known. In the case of finite-size DMRG, for example, for each system size it would just be a function of the truncated singular values, from which the error in the energy is straightforwardly calculated. That is why in the present method the authors must *guess* a proxy for the error in energy (Eq. (6) in the SI). So I do not entirely agree with the authors' statement in their reply that:

“we are not conflating the finite size error due to the wrong Hamiltonian (as used in finite cluster DMRG due to truncation of interactions) and the error due to lack of convergence of the wavefunction”.

It's true that the authors are studying the correct Hamiltonian, but any finite-size supercell will still contain what I would call “finite-size errors” in that the correlations are not properly taken into account at larger distances. So it seems to me that for any finite-size supercell they will have a lack of convergence of the wavefunction, i.e. the separation between wavefunction fidelity error and finite size error is not as clean as the authors would like.

That being said, once convergence is achieved in both hyperparameters, the physical result will be robust. And the authors seem to have demonstrated that in their work.

I would appreciate a short clarification of this point, so that readers (and future users of this algorithm) will have a better understanding of the assumptions made and of the calculation of the errors.

We thank the reviewers for their careful reading in the second round of review. Our understanding of the reviews is that Reviewers 2 and 3 are generally supportive of publication, while Reviewer 1 remains somewhat against. The comments by Reviewers 2 and 3 are mainly technical ones which we have now fully addressed by additional technical work and clarification in the manuscript. Reviewer 1’s position seems to center on only whether the new phase observed is interesting if it is not a long-range entangled phase (as opposed to any of the other new contributions, such as new methodology, new bulk phase diagram, reinterpretation of experiments); Reviewer 2 makes it clear that such a phase (even if not long-range entangled) is interesting. We have tried to further address the comments from Reviewer 1 about the new phase below.

We include each of the Reviewers’ comments in blue text, and our response to them inline with black text. In the manuscript, we have highlighted our new set of changes with magenta colored text.

1 Reviewer 1

Firstly, I appreciate the authors taking the time to respond to my comments and clarify various points in the manuscript. These changes have certainly improved the presentation of the paper. However, I stand by my assessment that the work is better suited for a more specialized journal than Nature Communications and continue to recommend against publication.

I understand that this is a subjective opinion, and I respect the authors’ right to maintain a different viewpoint, but let me mention four key reasons that inform my decision:

We appreciate Reviewer 1’s opinion that the manuscript has been improved and for a clear enumeration of their concerns.

1. As the authors themselves write, “However, given the finite size of all our calculations, we also cannot definitively rule out that when coarse grained over sufficiently long distances and in the TDL, the nematic phase is just a strong fluctuation around an entanglement stabilized crystalline state.” If the authors cannot establish that their “discovered” phase is truly a stable phase of matter in the thermodynamic limit, that dilutes one of the key results of this manuscript.

We note that the quote above does not appear in our manuscript, it is part of the previous response letter – in response to Reviewer 2, as to whether we are seeing a “cat” state. This is not related to the stability of the phase.

To be clear, we have established that the phase of matter is stable in the thermodynamic limit and it is a phase with strong fluctuations (i.e. entanglement) where the contributing configurations are *not* the classical crystal ground-state configurations. Instead it lives in the manifold of entanglement stabilized crystalline states, or in the terminology of Reviewer 2 that we have now extensively adopted in the current revision, configurations that emerge from an order from disorder mechanism. Such a state is not seen elsewhere in the phase diagram, and is novel. The novelty is also confirmed by Reviewer 2.

The only aspect that we cannot conclusively establish, is how long-range the entanglement is. It is clearly over the entire unit cell that we simulate, but it is likely that it is of finite range. This would require even larger simulations to confirm conclusively. Thus we adopted careful wording re: how long range the entanglement is, in both the response letter and in the paper, which may have been interpreted by the reviewer that we are not sure the phase is there, when in fact there is no doubt that this unusual phase is stable.

2. A generic quantum state is likely entangled to some degree. Just the presence of entanglement alone is not sufficient to make a state novel. The authors concede that “Whether or not there are additional long-range entanglement features is difficult to say”: satisfactorily answering this question is a prerequisite to demonstrate that the nematic state is an interesting phase of matter. If, as above, it is just a finite-size manifestation of “an entanglement stabilized crystalline state”, that would also appear entangled in numerical simulations despite being a rather uninteresting state.

We agree that all states are likely to have some amount of entanglement. The question here, which is essentially a matter of opinion, is whether the *only* interesting states in the physical world are long-range entangled states (which are extremely rare in Nature) or whether short-ranged entangled states are also interesting. We note that many phases long studied in condensed matter physics and that are considered interesting by a large community, such as the striped phases in the Hubbard model, are not known to show long-range entanglement. Instead, they are “unexpected” short-range entangled phases. This is exactly the kind of unexpected state that we find in the phase diagram.

In our original manuscript, we did not present a strong opinion as to whether the phase has short-range or long-range entanglement because, strictly speaking from the viewpoint of simulation, one would need further finite size scaling analysis and even larger simulations. In the current manuscript, based on the feedback from Reviewer 2, we have presented more of an opinion, that we believe the entanglement would ultimately likely be found to be “short-ranged” (although over quite a long distance).

3. The authors seem to agree in their response that the bulk nematic and the finite-size phase are not the same. This point is almost glossed over and rather subtly underemphasized in the sentence “the correlation functions reveal qualitative similarities to the bulk nematic phase”. If the finite-size phase is not even the true bulk entangled phase, then I fail to see why the properties of the idealized bulk phase are even relevant to experiments on finite systems.

We are not stating that the bulk nematic and finite-size phases are the same. We have tried to be extremely careful in our paper, which we feel is needed in this sort of problem where there are many competing phases, and we believe the reviewer may be mistaking our cautious wording for us not believing in our results! The fact is that the finite size fluctuations at the small size of 15×14 are strong, so it is difficult to talk about *any* phase (note that the boundary atoms themselves are almost 20 pct of the system!) It should be clear however, that the state does not have the correlations of the expected classical ground state. Instead the correlations are related to a superposition of the non-classical crystal ground states, similar to the composition of the bulk phase. In less precise terms, our language is meant to say that the state appears to be related to that in the bulk, but the phase is not “clean” at this system size due to boundary effects.

4. Beyond the observation of the nematic phase, all of the other results in the paper—such as the reinterpretation of the experimental results or the slight modification of phase boundaries—are, in my opinion, incremental improvements. I respect the numerical efforts that the authors have invested in Γ -point DMRG, which might be a useful technique. However, I do not believe that the development of a numerical method alone merits publication in Nature Communications unless accompanied by significant new results that were inaccessible previously.

As we have stated in our last response, in addition to the nematic phase, we feel it is important to obtain the correct results elsewhere in the phase diagram as well. Thus we respectfully disagree.

2 Reviewer 2

I thank the referees for addressing my points. The manuscript and the SI have been improved. I also still stand by the positive evaluation I gave in my first work: this numerical work sets an excellent standard for the booming field of many-body physics in Rydberg simulators, and this improved method

is evidenced by the authors discovering a phase of matter which had been overlooked in previous work, and which is stabilized by a different mechanism than the previously-identified ordered phases.

However, I strongly feel there are still two important issues that need to be resolved before I can recommend publication.

1) The first issue concerns the nematic phase. I appreciate the authors' extended discussion, both in their reply to my report, as well as the updated manuscript. However, I think certain aspects have not been accurately characterized:

First, the authors point out that the nematic states are not selected at the classical level but instead by a 4th order process in perturbation theory, and the authors call this an 'entanglement-stabilized phase'. I think I agree with this physical process, but it is dangerous/ill-advised to give a new name to a well-established process: I believe this is an instance of 'order from disorder'. In the quantum context, order from disorder works by starting with an extensively degenerate space of states (usually resulting from some classical interactions), and this degeneracy is then lifted by some additional quantum fluctuations (due to some classical/ordered states allowing for more fluctuations than other states). (E.g., see the discussion for the hexagonal lattice FFIM in [Moessner, Sondhi, Chandra, PRL 84, 4457 (2000)].) This general process seems to coincide with the mechanism found by the authors. We thank the reviewer for their general support of our work, and we agree that the phase we have found is interesting. We appreciate the reviewer pointing out this terminology. We have now used the terminology order-from-disorder in several places in the manuscript (e.g. in the abstract and main discussion.

Second, the authors say they are not sure whether the macroscopic entanglement they find in the nematic phase is a physically (un)stable (related to my question in my previous report). However, given the authors' extended discussion of this phase, it is now rather apparent to me that it is indeed unstable cat state entanglement, which can thus not extend to the thermodynamic limit. A characteristic property of cat state entanglement is that measuring, say, a single site, will collapse the global entanglement. This is clearly the case for the nematic state found by the authors. (E.g., a single-site measurement can reveal that a whole column is in the 'a' state.) Similarly, note that the "simple product state" the authors consider in the SI below Eq.8 is in fact a cat state along the vertical direction. If these points do not already convince the authors of the unphysical nature of their entangled state, let me add two more equivalent characterizations: if one were to re-do the numerical simulations with an offset chemical potential added to only *a single site* (which

favors an occupied site), the ground state will no longer have the observed entanglement; equivalently, one can find that the entangled state found by the authors has long-range order in a two-point function which does not satisfy the clustering/factorization property. It would be very unfortunate to publish a paper which can cause confusion (in the community) about the entangled nature of a state when the accurate characterization of its entanglement is within reach.

In conclusion, I believe the authors have discovered an ordered phase which is stabilized by 'order from disorder' in a Rydberg atom array. This phase is not characterized by a significant amount of entanglement, but it is still interesting as an instance of 'order from disorder'.

We believe that we and the reviewer are actually not far apart in how we understand the state. One question is whether the fluctuations/entanglement are truly long range (like a topologically ordered state) or whether they are strongly fluctuating non-classical crystals. Because of the finite size of the simulation, we would say that we cannot (purely from these computations) distinguish the two.

However, our *opinion* is also that unless long-range entanglement is definitively identified (e.g. from a topological entanglement entropy term) it is much more likely to be short-range entanglement. So we believe our view with respect to long-range entanglement is in complete accordance with the reviewer's view. We have therefore made it clear in the text that it is likely to be short-range entanglement.

The uncertainty that continues to persist (for us) is regarding the detailed nature of the state. The 4th-order perturbative process - which does not require a global shift of the columns - clarifies the stability of a *single* non-classical crystal via quantum fluctuations. However, our numerical results show equal participation in the ground state of the 6 equivalent permutations of this non-classical crystal, and strong fluctuations around these 6 to other varieties of crystals (with smaller wavefunction coefficients). Thus, there remains the possibility that the state we observe is vaguely analogous to the ground state of the square lattice AFM Heisenberg model, i.e. there are fluctuations around the *two* Neel states, rather than just one - in our case, the 6 equivalent crystals. We believe the reviewer is arguing that there should only be one, and the 6 are superimposed in a cat state. To examine this, we have carried out the finite field test. If we apply a finite field on a site (or set of sites), we find we need to apply a large field on at least two sites, of 10^{-3} or greater, to even see a small sign of partially collapsing the column state. This seems large for a pure cat state, as the entire energy scale separating classical and nonclassical crystals is only 10^{-4} per site. Normalizing the applied field to

the number of occupied sites in the classical crystal (roughly dividing by 10) yields a minimum field strength to see onset of collapse as roughly comparable to the scale of the perturbation stabilization – to us, a conclusive result would be to see that we require a much smaller field than this. Thus the accessible data seems to us to be inconclusive.

There may be a theoretical argument why only 1 crystal participates as the system size becomes very large. But as discussed above, our numerics do not make this clear. We believe larger calculations are needed to resolve this possibility. We do not make a strong case for this over the single crystal (we do not mention it in the main text, and it is only discussed as a possibility that is worth further study in the SI, section VIII) but we feel it is best not to conclusively rule out the “multiple Neel state” scenario, analogous to the Heisenberg AFM, with our current data and understanding.

Regardless, we have now made it clear in the text that we believe that the entanglement is likely short-ranged, and that we can then interpret this as an ordered state stabilized by strong, but short-range, entanglement in the order from disorder mechanism.

2) While I do not doubt the correctness of the authors’ Gamma-DMRG method, its discussion is sub-par.

First, I do not yet see why the authors’ definition of the Bloch basis is well-defined. The authors are considering an infinite tensor product Hilbert space $\mathcal{H} = \otimes \mathcal{H}_n$, where \mathcal{H}_n is a two-level qubit. The authors define the ‘Bloch’ states of the form $|\tilde{0}\rangle = |0\rangle + |L\rangle + |2L\rangle + \dots$. Note that these states are not defined in the many-body Hilbert space, nor are products of such states elements in the many-body Hilbert space. (Note that in contrast, in fermionic systems one can define Fourier operators/modes $c_k = c_x + c_{x+L} + \dots$ et cetera, and then define many-body states as $c_k|0\rangle$ where $|0\rangle$ is the many-body vacuum, which is a valid state in the many-body Hilbert space. Hardcore bosons do not allow for such a treatment.) The only way I (personally) can make sense of the authors definitions is by interpreting everything in a finite-dimensional Hilbert space of a $L \times L$ torus with periodic boundary conditions. If the authors do not wish to use such a torus geometry/Hilbert space, they have to include/discuss a more careful definition of their Hilbert space.

Second, in their reply to my report, the authors also agreed that the torus perspective is “also a natural perspective, although it is perhaps less clear how other observables are treated. However, we have included this point in the discussion given in the SI.” I cannot find a discussion of it in the SI (other than a discussion about periodicity of correlation functions). Also, it is not clear to me what the authors mean by “it is perhaps less clear how other

observables are treated”. Let me then rephrase my original question: if one were to perform torus DMRG (with interactions wrapping around rather than being truncated), would the simulations/code/results exactly coincide with what the authors have done? If yes: then why not state that more explicitly, and why give a new name to torus DMRG? If no: can the authors pinpoint a difference?

Let me state that if the authors’ method coincided with torus DMRG, I do not think that hurts the present work. Clearly the authors have used this tool in a novel way and shown it to be superior to previous treatments. But I do think it would be bad form to give a new name to an existing method, as it would very likely cause confusion in the literature. I thus encourage the authors to either clearly acknowledge and state that their method is torus DMRG, or pinpoint why it is not.

We thank the reviewer for these further insights into the source of confusion. We are happy to use the terminology torus DMRG with infinite wrap arounds, and we have now done so in the text. We have also retained the Gamma-DMRG description and expanded on the Hilbert space and other questions raised by the reviewer that are relevant in the SI.

We agree with the reviewer that taking the many-body Hilbert space, at a single point in the Brillouin zone, only *models* the infinite system – it does not actually generate a subspace of the Hilbert space of the infinite system. This is true in fermionic systems also: for example, for a 1-band tight-binding model at half-filling, with a single-site unit cell and Γ point boundary conditions, we only have one state $|1\rangle + |2\rangle + |3\rangle + \dots$, which is not a state in the many-body Hilbert space; nonetheless, we use expectation values of this state to say something about the infinite system. We have carefully clarified in the main text and SI that we are creating a model of the infinite system. This model Hilbert space is isomorphic to the torus Hilbert space.

We understand the reviewer finds the torus picture more intuitive, as may many people in the Rydberg community. We have therefore clarified in the main text and SI that what we are doing is equivalent to the torus model with infinite wrap-around. However, to give some motivation for why the model of the infinite system picture is natural to a different community, consider the evaluation of the interaction energy. If we replace the Rydberg interaction by $1/R$, then this is a Coulomb problem - and in the traditional evaluations of the Coulomb interaction energy of this type of problem, one considers this as a classical electrostatic problem defined in an infinite system, not the electrostatic problem of a torus with infinite wraparound. This leads to the ability to use a variety of procedures to perform the lattice sum, e.g. fast multipole methods, which sound strange in the torus language, where neighbouring cells

are not clearly identified. In fact, at least in our experience, in the electronic structure community, the torus language is always only identified with a cutoff potential, not a wrap around potential.

In any case, when appropriately defined, the two pictures are exactly equivalent. We thank the reviewer for pointing this out and encouraging us to be more explicit about this equivalence. Hopefully the additional text and explanation now allows us to address two audiences, who may find either terminology/picture more intuitive.

3 Reviewer 3

The authors have replied to all my comments in a satisfactory fashion.

I just have one more comment regarding the errors generated by the Gamma-point DMRG method. It is true, and now explained clearly in the SI, that there are two systematic sources of error in the method: supercell size and bond dimension. The authors have clearly explained that convergence in both is required, and have demonstrated such convergence numerically, thereby solidifying their infinite system phase diagram. However, I want to note that because of the constrained form of the entanglement allowed between supercells, the amount of entanglement neglected for a fixed supercell size is not known. In the case of finite-size DMRG, for example, for each system size it would just be a function of the truncated singular values, from which the error in the energy is straightforwardly calculated. That is why in the present method the authors must guess a proxy for the error in energy (Eq. (6) in the SI). So I do not entirely agree with the authors' statement in their reply that: "we are not conflating the finite size error due to the wrong Hamiltonian (as used in finite cluster DMRG due to truncation of interactions) and the error due to lack of convergence of the wavefunction". It's true that the authors are studying the correct Hamiltonian, but any finite-size supercell will still contain what I would call "finite-size errors" in that the correlations are not properly taken into account at larger distances. So it seems to me that for any finite-size supercell they will have a lack of convergence of the wavefunction, i.e. the separation between wavefunction fidelity error and finite size error is not as clean as the authors would like.

We thank the reviewer for the support for this work. The only question appears to be regarding a sentence in our response, which was a bit unclear. To clarify the sentence in our reply, the sentence should be read as "we are not conflating the finite size error due to the wrong Hamiltonian and the [finite size] error due to lack of convergence of the wavefunction [due to cell size]", i.e.

the lack of convergence of the wavefunction w.r.t. cell size is definitely a finite size error. Hopefully this clarifies our reply, which is in complete agreement with the reviewer. We have also slightly modified the wording in the SI to make it clearer.

That being said, once convergence is achieved in both hyperparameters, the physical result will be robust. And the authors seem to have demonstrated that in their work.

I would appreciate a short clarification of this point, so that readers (and future users of this algorithm) will have a better understanding of the assumptions made and of the calculation of the errors.

We have clarified the SI regarding this point.

REVIEWERS' COMMENTS

Reviewer #2 (Remarks to the Author):

I thank the authors for their detailed response, and for the revised manuscript. I like the new discussion in the manuscript of how this Rydberg system shows order-from-disorder. This is an exciting feature, which is clearly explained in this work. The additional significant entanglement is a bit more puzzling, but the authors now clearly state what they observe in their finite-size numerics, and they acknowledge that more future work is necessary to fully understand the fate of the entanglement in the thermodynamic limit. These combined findings are noteworthy and can inspire future work. And as I have already emphasized in my previous reports, the technical and detailed study is impressive, with a careful inclusion of long-range interactions. The authors have fully addressed my concerns, and I recommend this manuscript for publication in Nature Communications.